# DeepTheorem: Advancing LLM Reasoning for Theorem Proving Through Natural Language and Reinforcement Learning

## Abstract

Theorem proving serves as a major testbed for evaluating complex reasoning abilities in large language models (LLMs). However, traditional automated theorem proving (ATP) approaches rely heavily on *formal* proof systems that poorly align with LLMs' strength derived from *informal*, natural language knowledge acquired during pre-training. To fully leverage the theorem-proving knowledge acquired from pre-training, in this work, we present DeepTheorem, a comprehensive informal theorem-proving suite exploiting natural language to enhance LLM mathematical reasoning. DeepTheorem includes 1) **a large-scale dataset of 121K high-quality IMO-level informal theorems and proofs** spanning diverse mathematical domains, rigorously annotated for correctness, difficulty, and topic categories, accompanied by systematically constructed verifiable theorem variants; 2) **adaptation of RL-Zero explicitly to informal theorem proving**, leveraging the verified theorem variants to incentivize robust mathematical inference; 3) **comprehensive outcome and process evaluation metrics** examining proof correctness and the quality of reasoning steps; and 4) **a novel informal theorem proving benchmark** consoliadted from three established math competitions, formatted for automatic evaluation. Extensive experimental analyses demonstrate DeepTheorem significantly improves LLM theorem-proving performance compared to existing datasets and supervised fine-tuning protocols, achieving state-of-the-art accuracy and reasoning quality. Our findings highlight DeepTheorem's potential to fundamentally advance automated informal theorem proving and mathematical exploration.

## 1 Introduction

Theorem proving is widely regarded as a pinnacle challenge for evaluating advanced reasoning capabilities of both human and artificial intelligence. It requires integrating diverse cognitive facets such as abstraction, strategic inference, pattern recognition, and meticulous logical deduction. Recent advancements in deep learning, especially in large language models (LLMs), have significantly reshaped the landscape of automated theorem proving (ATP). Much prior work attempts ATP by integrating LLMs with either formal proof engines such as Lean, Coq, and Isabelle (Zheng et al., 2022; Liu et al., 2023; Tsoukalas et al., 2024) or domain-specific languages from ProofWiki (Welleck et al., 2022). However, these proof methods impose a significant barrier for LLMs whose primary strength derives from the vast corpus of natural language and LaTeX-based mathematical texts used during pre-training. This inherent misalignment limits LLMs' capability in theorem proving, leaving a considerable gap between their potential and actual performance.

In this paper, we present **DeepTheorem**, a novel, comprehensive suite expressly designed to leverage natural language to unleash the latent mathematical reasoning ability of LLMs for theorem proving. Instead of relying on formal proof assistants, DeepTheorem offers a scalable, intuitive, and flexible alternative, enabling LLMs to generate informal mathematical proofs aligning closely with human mathematicians' heuristic-driven thinking. Central to our approach is the construction of a large-scale benchmark comprising 121K IMO-level informal mathematical theorems with precise annotations such as correctness labels, difficulty levels, diverse mathematical domains, and verifiable theorem variants amenable to advanced reinforcement learning paradigms. Additionally,

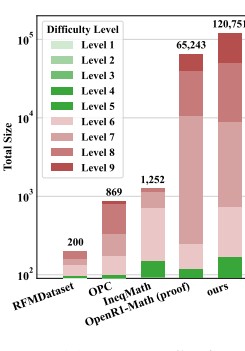 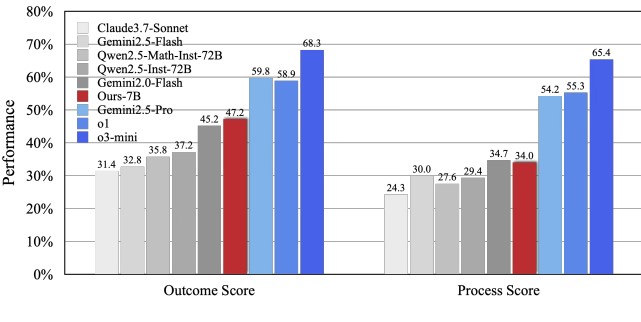

(a) Dataset Scale        (b) Performance

Figure 1: (a): Our dataset surpasses other informal theorem proving datasets in both scale and difficulty; (b): RL-Zero training with our DeepTheorem datasets on 7B model achieves strong results.

we adapt reinforcement learning algorithms to the problem of informal theorem proving for the first time, significantly enhancing LLM's ability to reason mathematically beyond the constraints of supervised fine-tuning (SFT). To evaluate existing and newly trained models on informal theorem proving, we also construct a new benchmark sourced from established mathematics competitions, and propose comprehensive evaluation metrics that rigorously assess the correctness of generated proofs and the processes underlying the proofs themselves.

Through extensive experiments, we show that leading LLMs still exhibit significant limitations in theorem proving. However, when trained with DeepTheorem, they achieve substantial performance improvements over models trained using existing datasets, showcasing the effectiveness of our natural-language-focused approach. Our results underscore the promise of DeepTheorem to redefine LLM-driven mathematical reasoning, offering a robust platform for continued progress in automated and scalable informal theorem proving.

To sum up, our key contributions are:

- We introduce the *DeepTheorem* framework, a comprehensive informal theorem-proving suite exploiting natural language to enhance LLM mathematical reasoning;

- We open-source a large-scale natural-language theorem collection of 121K informal mathematical theorems and corresponding high-quality proofs at IMO-level difficulty, suitable for both SFT and RL;

- We innovatively adapt the RL-Zero training method explicitly to informal theorem proving, significantly enhancing LLM's reasoning capacity beyond traditional SFT methods;

- We introduce a new benchmark for evaluating informal theorem proving, and develop a comprehensive evaluation framework assessing both the correctness of theorem proofs (outcome evaluation) and the completeness, logical validity, and correctness of generated reasoning processes (process evaluation);

- Through extensive experiments, we establish the superiority of our DeepTheorem training paradigm, achieving state-of-the-art performance and surpassing existing informal theorem datasets and training methods;

## 2 DATASET

**Overview** The *DeepTheorem* dataset[1] is a novel, large-scale resource designed to advance LLMs in informal mathematical theorem reasoning. Mined from a diverse web corpus, it addresses the need for challenging, decontaminated, and diverse topics to push LLMs toward frontier theorem proving. As illustrated in Figure 2, each entry in the dataset offers distinct features tailored to support diverse research objectives, including: 1) a mathematical theorem in standardized LaTeX format; 2) a True-or-False correctness label, where a False label indicates that the theorem can be mathematically disproved; 3) a fine-grained difficulty score ranging from level 5∼10; 4) mathematical topics for targeted analysis; and 5) detailed step-by-step proof solutions generated by o3-mini.

---

[1]In the rest of this section, *DeepTheorem* dataset refers specifically to the DeepTheorem training dataset.

**Question:** Let $f(z) = \frac{e^{iz}}{z}$, and consider the contour integral $\int_{\mathbb{R}} f(z)\,dz$ over the real line. Assume the contour is closed by a semicircular arc in the upper half-plane. Prove or disprove that the value of the improper integral $\int_{0}^{\infty} \frac{\sin x}{x}\,dx$ is given by: $\int_{0}^{\infty} \frac{\sin x}{x}\,dx = \frac{\pi}{2}$.

**Difficulty:** 8     **Final Answer: True**

**Topic:**
Calculus -> Integral Calculus -> Techniques Of Integration
Calculus -> Integral Calculus -> Applications Of Integrals
Algebra -> Intermediate Algebra -> Complex Numbers

**O3-mini Solution:** To prove this theorem, we start by writing
$$\int_{0}^{\infty} \frac{\sin x}{x}\,dx = \Im\left(\int_{0}^{\infty} \frac{e^{ix}}{x}\,dx\right)$$
Because the integrand $\frac{e^{iz}}{z}$ is meromorphic with a simple pole at $z = 0$, we consider the principal value of the integral along the whole real axis. That is, we consider
$$\text{p.v.} \int_{-\infty}^{\infty} \frac{e^{ix}}{x}\,dx$$
However, due to the pole at $z = 0$ lying on the real axis,

...

Thus, the theorem statement $\int_{0}^{\infty} \frac{\sin x}{x}\,dx = \frac{\pi}{2}$ has been proven.

boxed{True}

Figure 2: A data sample from the DeepTheorem dataset.

Comprising approximately 121K curated samples, *DeepTheorem* empowers large-scale exploration of LLM mathematical reasoning, curriculum learning, and cross-domain generalization. It serves as a versatile resource for advancing automated theorem proving, enhancing model reasoning capability, and developing adaptive learning frameworks, positioning it as a cornerstone for next-generation NLP research in mathematical theorem proving and logical reasoning. The key features of *DeepTheorem* are:

- **Large scale**: Unlike other informal theorem proving datasets that focus on small-scale studies of specific domains (Sheng et al., 2025) or analyzing LLM-generated proofs (Dekoninck et al., 2025; Guo et al., 2025), *DeepTheorem* dataset leverages the vastness of web-sourced content to construct a training set, ensuring comprehensive coverage of mathematical concepts and problem types. As shown in Figure 1(a), our dataset consists of approximately 121K theorems, significantly outscaling prior datasets. The only dataset comparable in size to *DeepTheorem* is OpenR1-Math (Face, 2025). However, this dataset is not explicitly annotated for mathematical theorem proving and thus not readily usable for training theorem-proving models.

- **Frontier and extremely challenging theorems**: *DeepTheorem* dataset is distinguished by its inclusion of advanced mathematical theorems, each annotated with difficulty levels to enable targeted evaluation and training

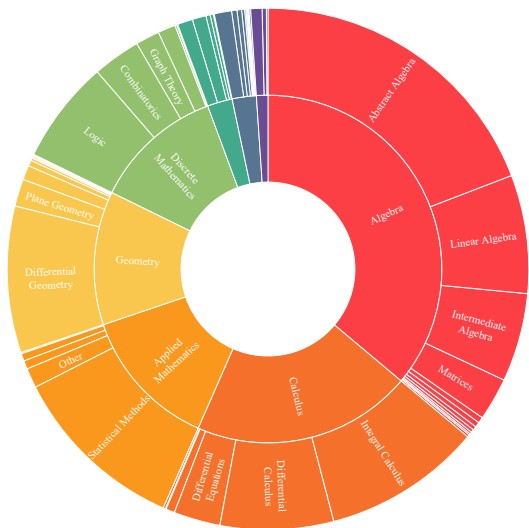

Figure 3: Statistics of *DeepTheorem* dataset hierarchical topics. See Appendix A for details.

across a spectrum of complexities. As shown in Figure 4, *DeepTheorem* dataset emphasizes theorems at high difficulty levels (6–9), surpassing existing corpora in complexity and challenge, presenting significant challenges for state-of-the-art LLMs while aligning with frontier, IMO-level benchmarks such as FIMO (Liu et al., 2023).

- **Diverse topics**: As shown in Figure 3, *DeepTheorem* dataset captures the breadth of informal theorem-based reasoning by covering nearly the entirety of the mathematical landscape, including algebra, discrete math, applied math, calculus, geometry, mathematical analysis, number theory, etc. By encompassing this wide array of domains, *DeepTheorem* dataset enables researchers to assess model performance on both specialized and interdisciplinary mathematical tasks, fostering the development of LLMs that can generalize effectively across the full spectrum of mathematical reasoning.

- **Strict decontamination**: To preserve evaluation integrity, DeepTheorem dataset employs rigorous decontamination processes to avoid overlap with widely used benchmarks. The targets of

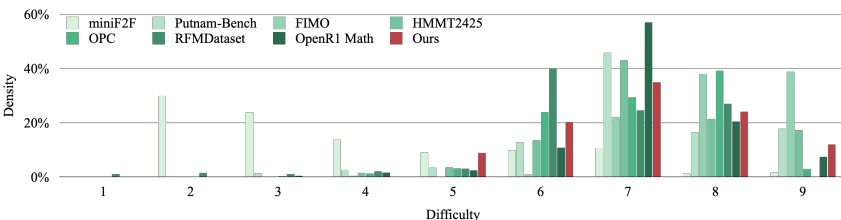

Figure 4: Difficulty density histogram comparison across widely used benchmarks (i.e. miniF2F, Putnam-Bench, FIMO, HMMT2425) and other recent theorem proving datasets (i.e. OPC, RFM-Dataset, theorem subset of OpenR1-Math).

our decontamination includes general math reasoning benchmarks - MATH (Hendrycks et al., 2021b), AIME (MAA, a), AMC (MAA, b), Minerva Math (Lewkowycz et al., 2022), Olympiad-Bench (He et al., 2024), Omni-MATH (Gao et al., 2025), MathOdyssey (Fang et al., 2024), GAOKAO (Zhong et al., 2024), JEEBench (Arora et al., 2023), MMLU-STEM (Hendrycks et al., 2021a), CMATH (Wei et al., 2023), OlympicArena (Huang et al., 2024), GSM8K (Cobbe et al., 2021), GPQA (Rein et al., 2024) - and theorem proving benchmarks: miniF2F (Zheng et al., 2022), PutnamBench (Tsoukalas et al., 2024), FIMO (Liu et al., 2023), and HMMT (Harvard-MIT Mathematics Tournament, 2024, 2025).

- **Proofs from advanced LLMs**: *DeepTheorem* dataset includes concise, high-quality proof solutions generated by o3-mini, tailored for supervised fine-tuning (SFT). These proofs provide a compact yet complete outline of the logical steps required to prove (or disprove) each theorem, optimized for clarity and brevity. Unlike verbose or overly formal proofs, these proofs, expressed in LaTeX, align with the informal nature of LLMs, making them an effective learning signal. By incorporating these proofs, the dataset enables models to internalize structured reasoning patterns, improving their ability to generate coherent and logically sound mathematical arguments.

## 2.1 CONSTRUCTING *DeepTheorem* DATASET

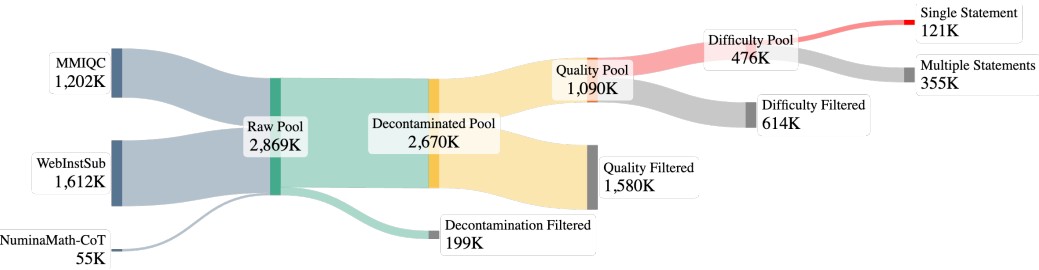

Figure 5: The process pipeline of constructing *DeepTheorem* dataset.

As shown in Figure 5, the construction pipeline of *DeepTheorem* dataset starts by aggregating raw data from multiple sources, including MMIQC (Liu et al., 2025), WebInstruct (Yue et al., 2024), and NuminaMath-CoT (LI et al., 2024).

**Decontamination** To ensure the integrity of *DeepTheorem* dataset, we implement a rigorous decontamination process to eliminate overlap between training and testing datasets. The process employs a recall-and-justify pipeline to identify and remove potential duplicates, consisting of three key steps:

1. *Embedding Generation*: We use an embedding model[2] (Reimers & Gurevych, 2019; Toshniwal et al., 2025) to generate sentence embeddings for all theorem statements in the training and testing datasets.

---

[2] `paraphrase-multilingual-MiniLM-L12-v2`

2. *Similarity Recall*: For each training sample, we compute its embedding cosine similarity to all test samples, and recall the top five testing samples exceeding a similarity threshold (set to 0.7).

3. *Contamination Justification*: An LLM[3] evaluates whether the recalled test samples are contaminated within the current training sample (Section F.1).

This process removed approximately 199K contaminated samples, effectively identifying identical cases, generalized questions, and converse theorems, Removed contaminated examples are shown in Appendix B. Approximately 2.6M samples remain for the next processing step.

**Quality control and proof generation** We also implement a rigorous quality control pipeline for generating and validating theorem statements and their proofs. The process involves four key steps:

1. *Theorem Justification*: An LLM verifies that the question is complete, and indeed a theorem-proving question (Section F.2).

2. *Rationale Summarization*: An LLM summarizes the original question and generates a formatted, concise, self-contained theorem (Section F.3).

3. *Proof Generation*: o3-mini (high effort) generates the proof solution with True-or-False conclusion about the theorem (Section F.4).

4. *Logical Validation*: The LLM performs an extra justification step to check that the theorem-proof pair is logically coherent. (Section F.5)

This systematic approach yielded 1.08M high-quality, mathematically sound theorem-proof pairs.

**Difficulty and single statement annotation** We annotate the difficulty levels of *DeepTheorem* dataset , and remove questions with multiple statements to prove.

1. *Difficulty Annotation*: An LLM analyzes each theorem statement following the strategy of Gao et al. (2025), considering factors such as logical complexity, mathematical prerequisites, and proof length, to assign a difficulty score on a scale of 1 to 9. Only questions with a difficulty score of at least 5 are retained. (Examples in Section B, prompt in Section F.6)

2. *Single-Statement Filtering*: We filter out samples that query for proving multiple statements, retaining only those with a single, well-defined theorem to ensure clarity and consistency with evaluation.

3. *Topic Annotation*: Finally, we annotate the topic domain of the mathematical theorems with LLMs. (Section F.7)

The difficulty and single statements filtering results in 121K challenging theorems, yielding the final *DeepTheorem* dataset.

## 3 THEOREM PROVING VIA REINFORCEMENT LEARNING

**Motivation** Conventionally, informal theorem-proving datasets are utilized through supervised fine-tuning (SFT), where models learn to generate proofs by imitating dataset examples. However, recent studies on RL-Zero demonstrate its superior performance over SFT by leveraging a base model's pretrained knowledge and exploratory capabilities (Jaech et al., 2024; DeepSeek-AI et al., 2025). This raises a natural question: *Can we harness the base model's exploration ability for informal theorem proving?* In this section, we explore the possibility of utilizing RL-Zero for informal theorem proving. The process involves three key steps: 1) data augmentation to generate contradictory theorem variants for binary rewards; 2) RL-Zero training with GRPO (Shao et al., 2024); and 3) Evaluation of the theorem-proof generation.

### 3.1 THEOREMS WITH VERIFIABLE REWARDS

**Theorems can be disproved** To construct a theorem with rewards for RL-Zero, we make the key observation:

---

[3] GPT-4o is used for annotation in this section unless otherwise specified.

> **Insight**
>
> A statement need not be correct but can be also proven incorrect, enabling a binary reward structure compatible with RL-Zero.

This observation allows us to transform *DeepTheorem*'s theorems into true-or-false variants, facilitating RL training that incentivizes robust reasoning.

To construct such training data, we use an LLM to expand the original theorems into contradictory variants that can be *disproved*. Specifically, we strictly limit the transformation made to the original theorem, so that the resulting variant is either entailed by or contradictory from the original theorem. Consider the example in Table 1 (omitting the hypotheses for simplicity): if the original theorem can be proved, Variant #1 is also correct and can be mathematically proved in the same manner as the original one, while Variant #2 must be incorrect and can be disproved.

Table 1: An example of theorem variants given an original theorem.

| Theorems | Example |
|----------|---------|
| Original | $x > 1$ |
| Variant 1 | $x > 0$ |
| Variant 2 | $x < 1$ |

With such logically entailing or contradictory transformations, we are able to construct variants of a theorem that are guaranteed to be correct or incorrect by only accessing the theorem itself but not the proof process, which makes this transformation task much easier than annotating new math statements, and thus allowing a relatively weaker LLM (e.g. Qwen2.5-72B-Instruct, Yang et al., 2024) to perform it. After this expansion phase, we further annotate the completeness of the resulting theorem pool and finally acquire a training set of 242K mathematical theorems that can either be proved or disproved, each with a complete proof trajectory (see Appendix F.8 for more details).

### 3.2 BINARY REWARDS ACTIVATE THEOREM PROOF GENERATION

With the aforementioned theorem variants, we can now apply reinforcement learning to natural language theorem proving. Specifically, we adopt the GRPO algorithm (Shao et al., 2024).

**Proof generation with RL**   Inspired by the success of reasoning-specialized models such as R1 and its open-source reproductions (DeepSeek-AI et al., 2025; Hu et al., 2025), we encourage the model to enclose its reasoning process in <think> </think> tags in the system prompt to incentivize more detailed reasoning behaviours (see Appendix F.9), and then ask the model to end each proof with either "\boxed{proved}" or "\boxed{disproved}". In the reward function, we extract this answer and compare it against the ground truth, giving a reward of 1 if the answer matches, and 0 otherwise. We also enforce several sanity checks to prevent model collapse: if the ratio of white spaces in a model's solution is less than 0.05 or the average character repetition count is greater than 300, then a reward of 0 is issued regardless of the answer.

### 3.3 EVALUATION

The theorem-proving questions used for evaluation are drawn from two challenging benchmarks — FIMO (Liu et al., 2023) and Putnam (Tsoukalas et al., 2024) — and a newly constructed theorem-proving subset of HMMT (Harvard-MIT Mathematics Tournament, 2024, 2025).

**Outcome evaluation**   Evaluating the correctness of natural language (NL) proofs poses a significant challenge, as it mirrors the complexity faced by humans in assessing the logical coherence and mathematical validity of informal reasoning. Unlike formal theorem-proving systems that rely on structured logic, NL proofs lack a standardized format, making their evaluation inherently subjective and difficult to automate. To address this, we propose a novel evaluation framework that leverages multiple en-

Table 2: Test data statistics. Each original theorem is manually expanded into multiple entailing or contradictory variants. *Random accuracy* indicates the expected score of random guessing following the outcome criteria described below.

| Bench | Scale | Variants (Avg.) | Random Acc. |
|-------|-------|-----------------|-------------|
| FIMO | 172 | 2.7 | 17.4 |
| HMMT | 205 | 3.5 | 11.2 |
| Putnam | 281 | 2.9 | 15.4 |

tailing and contradictory variants derived from each theorem. By assessing the model's ability to consistently assign correct truth values across these variants, we indirectly estimate its theorem justification ability. When the number of variants is sufficiently large, this approach provides a robust proxy for evaluating the correctness of NL proof generation.

Thus, we manually expand each question in the three data sources into multiple entailing or contradictory variants following the same variant generation protocol in Section 3.1, and the resulting benchmarks are shown in Table 2. When evaluating a model, we ask it to either prove or disprove each theorem and corresponding variants, and evaluate the results with the criteria below:

---

**Outcome Criteria**

A test case in a theorem testing set is passed if and only if:

1. The model explicitly produces a truth value (true or false) for theorems and variants;
2. The predicted truth value for the original theorem is correct;
3. The predicted truth values for all entailing variants are the same as the original theorem;
4. The predicted truth values for all contradictory variants are the inverse of the original theorem.

---

**Process evaluation**   Since theorem proving requires generating logically validated proofs for each reasoning step, we also develop a process evaluation framework that evaluates the quality of proof along four dimensions:

- **Logical Validity**: Check if each step follows logically from the previous one. Flag any logical errors.;
- **Completeness**: Verify if all necessary cases and steps are included to prove the theorem;
- **Correctness**: Confirm if the final conclusion is correct;
- **Clarity**: Assess if the proof is clear, unambiguous, and well-explained.

We use GPT-4o as the LLM judge and ask it to score the proof using a weighted sum of the four dimensions (prompt given in Appendix F.10). In Appendix D, we also present the results using o3-mini and Ling-1T (Team & AI, 2025) as the judge as well as human evaluation.

## 4   EXPERIMENTS

### 4.1   SETTINGS

We train two sets of models, using supervised fine-tuning (SFT) and zero reinforcement learning (RL-Zero) respectively, starting from Qwen2.5-Base (Yang et al., 2024) (additional results with Qwen3-Base (Yang et al., 2025) are provided in Appendix C). For SFT, we train the models for 3 epochs on the complete proof solutions in the dataset, using one machine for training each model. For RL-Zero, we adopt GRPO with batch size 128, group size 64, and maximum rollout length 8192. We train the models for 1000 steps, and distribute each model across two machines during training. Following the settings of Hu et al. (2025), we do not apply any KL regularization or entropy loss, as we find that KL regularization has a negligible impact on model performance, while entropy loss leads to model collapse.

As a baseline, we select the theorem-proving subset of OpenR1-Math (Face, 2025), the highest-quality existing theorem-proving dataset with complete questions and responses. We apply the same processing pipelines to it as detailed in Section 2.1, which yields 66K original theorems and 130K variants in total. We dub this processed dataset *OpenR1-Math-Proof*.

### 4.2   MAIN RESULTS

***DeepTheorem* with RL-Zero achieves the best performance**   The main results are presented in Table 3. *DeepTheorem* demonstrates superior performance over OpenR1-Math-Proof, especially

Table 3: Outcome (out.) and Process (proc.) evaluation of models trained on OpenR1-Math-Proof and DeepTheorem.

| Model | Strategy | Data | FIMO | | HMMT | | Putnam | | Avg. | |
|---|---|---|---|---|---|---|---|---|---|---|
| | | | *out.* | *proc.* | *out.* | *proc.* | *out.* | *proc.* | *out.* | *proc.* |
| **1.5B** | SFT | OpenR1-Proof | 20.63 | 8.66 | 11.86 | 4.80 | 35.42 | 18.98 | 22.64 | 10.81 |
| | | DeepTheorem | 31.75 | **18.86** | 15.25 | 9.41 | 36.46 | 21.43 | 27.82 | **16.57** |
| | RL | OpenR1-Proof | **34.92** | 8.54 | 16.95 | 5.10 | **55.21** | 17.92 | 35.69 | 10.52 |
| | | DeepTheorem | 31.75 | 15.23 | **23.73** | **10.15** | 52.08 | **22.79** | **35.85** | 16.06 |
| **3B** | SFT | OpenR1-Proof | 23.81 | 12.85 | 15.25 | 6.90 | 43.75 | 27.96 | 27.60 | 15.90 |
| | | DeepTheorem | 33.33 | 20.38 | 20.34 | 12.15 | 36.46 | 25.43 | 30.04 | 19.32 |
| | RL | OpenR1-Proof | 34.92 | 14.33 | 23.73 | 11.72 | **57.29** | **35.11** | **38.65** | 20.39 |
| | | DeepTheorem | **38.10** | **23.39** | **25.42** | **13.56** | 52.08 | 33.84 | 38.53 | **23.60** |
| **7B** | SFT | OpenR1-Proof | 30.16 | 18.23 | 15.25 | 8.63 | 48.96 | 32.95 | 31.46 | 19.94 |
| | | DeepTheorem | 34.92 | 26.69 | 22.03 | 15.41 | 41.67 | 33.50 | 32.87 | 25.20 |
| | RL | OpenR1-Proof | 42.86 | 22.79 | 25.42 | 13.15 | **60.42** | 38.94 | 42.90 | 24.96 |
| | | DeepTheorem | **55.56** | **39.07** | **28.81** | **20.85** | 57.29 | **42.20** | **47.22** | **34.04** |

for the 7B backbone and in terms of process evaluation. On the other hand, our RL-Zero training paradigm consistently outperforms SFT, validating the effectiveness of RL-Zero in pushing the models' reasoning capabilities beyond the limit of SFT.

**DeepTheorem achieves strong parameter efficiency** We demonstrates that our DeepTheorem-RL strategy achieves strong parameter efficiency in Figure 6. Compared to the Qwen2.5 series, training DeepTheorem on 1.5 to 7B models significantly improves the informal theorem proving boundary at parameter-performance space. Moreover, when extrapolated DeepTheorem parameter efficiency also surpasses SOTA commercial models such as o1 and o3-mini.

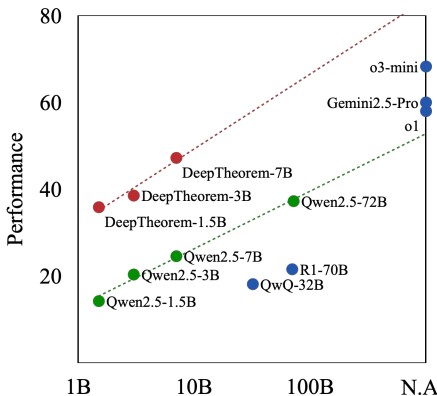

Figure 6: Performance improvement of models trained with DeepTheorem over baselines on theorem proving benchmarks.

**SOTA performance at equal model scale** In Table 4, we also provide the evaluation results of SOTA LLMs on the three benchmarks. These results suggest that theorem proving, especially our newly constructed HMMT benchmark, is still quite challenging for LLMs. On the other hand, our 7B model, trained with RL-Zero on *DeepTheorem*, outperforms SOTA models of much larger sizes, including those specialized in math and reasoning, demonstrating the superior quality of *DeepTheorem* and our innovative outcome-supervised RL training approach for theorem proving.

**Reasoning with theorem proving skills** In Figure 7, we visualize the techniques used by our 7B model trained with RL on *DeepTheorem*, where direct proof is most commonly used, followed by proof by exhaustion and construction. In Appendix E, we provide a non-cherry-picked example generation, finding the model to deliver a clear and correct disproof, highlighting its efficacy in tackling advanced mathematical problems with precision and clarity.

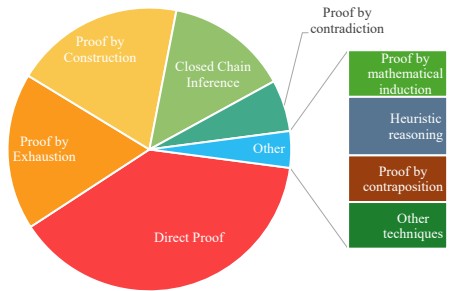

Figure 7: Distribution of proof techniques used by DeepTheorem-7B.

Table 4: Evaluation comparison of our model trained with RL on DeepTheorem with SOTA LLMs for both commercial models and open source models. *: denotes our method. Inst: Instruct; DS: Deepseek.

| Model | FIMO | | HMMT | | Putnam | | Avg. | | #Rank | |
|-------|------|------|------|------|--------|------|------|------|-------|------|
| | *out.* | *proc.* | *out.* | *proc.* | *out.* | *proc.* | *out.* | *proc.* | *out.* | *proc.* |
| *Commercial Models* | | | | | | | | | | |
| Claude3.7-Sonnet | 34.92 | 26.28 | 13.56 | 8.29 | 45.83 | 38.33 | 31.44 | 24.30 | 12 | 12 |
| Gemini2.5-Pro | 57.14 | 54.06 | 57.63 | 49.82 | 64.58 | 58.75 | 59.78 | 54.21 | 4 | 4 |
| Gemini2.5-Flash | 30.16 | 28.95 | 25.42 | 22.02 | 42.71 | 38.98 | 32.76 | 29.98 | 11 | 8 |
| GPT-4o | 34.92 | 30.70 | 16.95 | 14.59 | 22.92 | 18.88 | 24.93 | 21.39 | 14 | 13 |
| o1-mini | 60.32 | 55.23 | 35.59 | 30.90 | 61.46 | 52.88 | 52.46 | 46.34 | 6 | 6 |
| o1 | 66.67 | 61.00 | 47.46 | 47.30 | 62.50 | 57.55 | 58.88 | 55.28 | 5 | 3 |
| o3-mini | 80.95 | 77.61 | 45.76 | 43.47 | 78.12 | 75.12 | 68.28 | 65.40 | 2 | 1 |
| *Open Source Models* | | | | | | | | | | |
| Qwen2.5-Inst-7B | 30.16 | 21.13 | 10.17 | 6.83 | 33.33 | 25.39 | 24.55 | 17.78 | 15 | 15 |
| Qwen2.5-Inst-72B | 49.21 | 37.35 | 13.56 | 9.78 | 48.96 | 41.00 | 37.24 | 29.38 | 8 | 9 |
| Qwen2.5-Math-Inst-7B | 28.57 | 18.86 | 3.39 | 1.61 | 25.00 | 18.79 | 18.99 | 13.09 | 17 | 18 |
| Qwen2.5-Math-Inst-72B | 47.62 | 36.02 | 11.86 | 8.61 | 47.92 | 38.04 | 35.80 | 27.56 | 10 | 11 |
| R1-Distill-7B | 6.35 | 4.27 | 0.00 | 0.00 | 4.17 | 2.58 | 3.51 | 2.28 | 19 | 19 |
| R1-Distill-70B | 17.46 | 14.05 | 16.95 | 13.52 | 30.21 | 23.10 | 21.54 | 16.89 | 16 | 16 |
| QwQ-32B | 17.46 | 15.41 | 11.86 | 10.10 | 25.00 | 18.19 | 18.11 | 14.57 | 18 | 17 |
| Llama3.3-Inst-70B | 41.27 | 27.33 | 10.17 | 4.12 | 36.46 | 25.30 | 29.30 | 18.92 | 13 | 14 |
| Qwen3-32B | 73.02 | 55.04 | 52.54 | 36.88 | 79.17 | 57.81 | 68.24 | 49.91 | 3 | 5 |
| Kimi-K2-Thinking-1T | 25.40 | 19.49 | 32.20 | 25.77 | 51.04 | 40.40 | 36.21 | 28.56 | 9 | 10 |
| Ling-1T | 85.71 | 64.27 | 55.93 | 37.09 | 83.33 | 64.79 | 74.99 | 55.38 | 1 | 2 |
| *DeepTheorem-RL-7B | 55.56 | 39.07 | 28.81 | 20.85 | 57.29 | 42.20 | 47.22 | 34.04 | 7 | 7 |

## 5 RELATED WORK

**Theorem proving**   Theorem proving is a non-trivial task for natural language processing. In the context of LLMs, most works on theorem proving focus on formal languages such as Lean, Coq, and Isabelle (Xin et al., 2024; Ren et al., 2025). NaturalProofs (Welleck et al., 2021) and NaturalProver (Welleck et al., 2022) represent pioneering works that attend to informal theorem proving, though utilizing a domain-specific language from the ProofWiki website rather than the more human-accessible LaTeX-based natural language. Concurrent to our work, several small-scale informal theorem proving datasets have been introduced to the community, including OPC (Dekoninck et al., 2025), IneqMath (Sheng et al., 2025), and RFMDataset (Guo et al., 2025). However, these datasets are designed for analyzing LLMs' reasoning errors and failure modes in theorem proving, whereas *DeepTheorem* represents the first attempt at scaling up the training of informal theorem proving models.

**Learning to reason with RL-zero**   RL-Zero (DeepSeek-AI et al., 2025) is a streamlined framework designed to develop reinforcement learning capabilities in LLMs without SFT. While recent advances in LLM reasoning have been significantly influenced by RL techniques (Jaech et al., 2024; DeepSeek-AI et al., 2025; Team, 2024; xAI, 2025; Google, 2025), existing approaches predominantly focus on closed-form questions, addressing only a subset of reasoning problems. In contrast, we investigate the application of RL-Zero in process-oriented reasoning, specifically in informal theorem-proving. To the best of our knowledge, this is the first study to apply RL-Zero to informal theorem proving, marking a significant advance in enabling LLMs to address more diversified reasoning tasks in mathematical and logical domains.

## 6 CONCLUSION

In this paper, we introduce DeepTheorem, a novel comprehensive theorem-proving suite involving a large-scale annotated dataset of 121K IMO-level informal mathematical theorems and corresponding high-quality natural-language proofs, alongside systematically constructed verifiable theorem variants. We further adapt RL-Zero method to informal theorem reasoning, significantly surpassing supervised fine-tuning in performance. Comprehensive evaluations involving outcome accuracy and detailed process assessment on our newly constructed benchmark demonstrate the effectiveness

of our approach, achieving state-of-the-art theorem-proving performance and significantly pushing LLM reasoning boundaries. Through these contributions, DeepTheorem provides a robust foundation for future advancements in automated mathematical theorem proving, leveraging natural language flexibility to empower scalable, human-like reasoning abilities in large language models.

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

# A    TOPIC DISTRIBUTION IN *DeepTheorem* DATASET

Table 5: Subject and topic distribution in the *DeepTheorem* dataset.

| Subject | Topic | Frequency |
|---|---|---|
| Algebra | Abstract Algebra | 23134 |
| | Linear Algebra | 8858 |
| | Intermediate Algebra | 6634 |
| | Matrices | 3201 |
| | Sequences And Series | 440 |
| | Vectors | 383 |
| | Prealgebra | 281 |
| | Algebraic Expressions And Inequalities | 226 |
| | Other | 448 |
| Calculus | Integral Calculus | 11844 |
| | Differential Calculus | 8478 |
| | Differential Equations | 3478 |
| | Limits | 664 |
| | Other | 232 |
| Applied Mathematics | Statistical Methods | 13199 |
| | Probability | 632 |
| | Math Word Problems | 557 |
| | Other | 1583 |
| Geometry | Differential Geometry | 10951 |
| | Plane Geometry | 2044 |
| | Solid Geometry | 1082 |
| | Non Euclidean Geometry | 433 |
| | Other | 504 |
| Discrete Mathematics | Logic | 7710 |
| | Combinatorics | 3590 |
| | Graph Theory | 1790 |
| | Algorithms | 1318 |
| | Other | 225 |
| Number Theory | Prime Numbers | 1134 |
| | Congruences | 1032 |
| | Factorization | 318 |
| | Other | 315 |
| Mathematical Analysis | Calculus | 407 |
| | Applied Mathematics | 338 |
| | Geometry | 221 |
| | Other | 1746 |
| Other | Topology | 317 |
| | Other | 1007 |

## B  EXAMPLES OF SAMPLE DIFFICULTIES AND DECONTAMINATED TRAINING CASES

Table 6: Example questions in *DeepTheorem* dataset from different difficulty levels.

| Level | Example | Topic |
|---|---|---|
| 5 | Let $A$ be a commutative ring with identity, and let $a, b \in A$ where $a$ is in the Jacobson radical of $A$. Prove or disprove that the element $1 - ab$ is a unit in $A$, i.e., the principal ideal $(1 - ab)$ generated by $1 - ab$ is equal to $A$. | Abstract Algebra - Ring Theory |
| 6 | Let $M$ be a topological manifold with an open cover $\{U_i\}$ such that each $U_i$ is diffeomorphic to an open subset of $\mathbb{R}^n$. Prove or disprove that if each $U_i$ is diffeomorphic to the open unit ball in $\mathbb{R}^n$, then $\{U_i\}$ forms a contractible open cover of $M$. | Differential Geometry - Manifolds |
| 7 | Let $\mathbf{N}^*$ denote the set of all positive integers. Suppose $a > 0$ is a real number, and there exist $n$ mutually disjoint infinite subsets $A_1, A_2, \ldots, A_n$ such that $A_1 \cup A_2 \cup \cdots \cup A_n = \mathbf{N}^*$, and for each $i = 1, 2, \ldots, n$, and any $b, c$ in $A_i$, where $b > c$, the condition $b - c \geq a^i$ holds. Prove or disprove that $a$ must satisfy $0 < a < 2$. | Intermediate Algebra - Polynomial Operations |
| 8 | Let $f(z) = \frac{e^{iz}}{z}$, and consider the contour integral $\int_{\mathbb{R}} f(z)\,dz$ over the real line. Assume the contour is closed by a semicircular arc in the upper half-plane. Prove or disprove that the value of the improper integral $\int_0^\infty \frac{\sin x}{x}\,dx$ is given by: $\int_0^\infty \frac{\sin x}{x}\,dx = \frac{\pi}{2}$. | Integral Calculus - Techniques of Integration |
| 9 | Let $F_3(a, b_1, b_2, c_1, c_2; x, y)$ denote Appell's hypergeometric function of two variables. For parameters $\alpha, \beta, \gamma, \nu, \rho \in \mathbb{C}$ such that $\mathrm{Re}(\gamma) > 0$ and $\mathrm{Re}(\nu - \gamma) > 0$, prove or disprove that the integral $$\mathcal{I}(\alpha, \beta, \gamma, z; \gamma, \nu, \rho, w) = \int_0^1 t^{\gamma-1}(1-t)^{\nu-\gamma-1}(1-zt)^{-\alpha}(1-wt)^{-\beta}\,dt$$ evaluates to $$\mathcal{I}(\alpha, \beta, \gamma, z; \gamma, \nu, \rho, w) = \frac{B(\gamma, \nu - \gamma)}{(1-w)^\rho} F_3\left(\rho, \alpha, \nu - \gamma, \beta, \nu; \frac{w}{w-1}, z\right),$$ where $B(x, y)$ is the Beta function. | Integral Calculus - Multi Variable |

Table 7: Examples of benchmark contamination in polynomial and number theory problems. Generalizing and logically equivalent parts are highlighted.

| Contaminated Example | Benchmark Example | Relationship |
|---|---|---|
| Let $p(x)$ be a univariate polynomial. Then $p(x)$ is nonnegative for all $x \in \mathbb{R}$ if and only if $p(x)$ can be expressed as a sum of squares (SOS), i.e., $p(x) = \sum_{i=1}^k q_i^2(x)$ for some polynomials $q_1(x), \ldots, q_k(x)$. | Let $p(x)$ be a polynomial that is nonnegative for all real $x$. Prove that for some $k$, there are polynomials $f_1(x), \ldots, f_k(x)$ such that $p(x) = \sum_{j=1}^k (f_j(x))^2$. | Identical |
| Let $p(x_1, x_2, \ldots, x_n)$ be a real polynomial. If $p(x_1, x_2, \ldots, x_n)$ is non-negative for all $(x_1, x_2, \ldots, x_n) \in \mathbb{R}^n$, then $p(x_1, x_2, \ldots, x_n)$ can be expressed as a sum of squares of polynomials if and only if $p$ belongs to the quadratic module generated by the constraints of a certain semialgebraic set. Formally, there exists a set of polynomials $q_i(x_1, x_2, \ldots, x_n)$ such that $p(x_1, x_2, \ldots, x_n) = \sum_{i=1}^k q_i(x_1, x_2, \ldots, x_n)^2$, provided certain conditions on $p$ and the domain hold to ensure the SOS representation. | Let $p(x)$ be a polynomial that is nonnegative for all real $x$. Prove that for some $k$, there are polynomials $f_1(x), \ldots, f_k(x)$ such that $p(x) = \sum_{j=1}^k (f_j(x))^2$. | Generalizing |
| Let $n$ be a positive integer. If $n$ is not prime, then $2^n - 1$ is not prime. | Show that if $n$ is a positive integer and $2^n - 1$ is prime, then $n$ is prime. | Logically Converse |

# C  ADDITIONAL EXPERIMENTS

Table 8: Performance comparison and generation length of Qwen3-4B trained on different data, all sampled to 100K size.

| Strategy | Data | FIMO | | HMMT | | Putnam | | Avg. | | Gen. Length |
|---|---|---|---|---|---|---|---|---|---|---|
| | | *out.* | *proc.* | *out.* | *proc.* | *out.* | *proc.* | *out.* | *proc.* | |
| SFT | OpenR1-Proof | 25.40 | 9.07 | 15.25 | 2.43 | 44.79 | 11.07 | 28.48 | 7.53 | 1572 |
| | DeepTheorem$_{easy}$ | 30.16 | 10.07 | 22.03 | 9.10 | 36.46 | 13.30 | **29.55** | 10.82 | 1305 |
| | DeepTheorem$_{hard}$ | 30.16 | 13.09 | 15.25 | 4.35 | 40.62 | 16.08 | 28.68 | **11.17** | 1443 |
| | DeepTheorem | 28.57 | 11.83 | 15.25 | 3.68 | 39.58 | 15.44 | 27.80 | 10.31 | 1266 |
| RL | OpenR1-Proof | 41.27 | 14.16 | 25.42 | 8.32 | 55.21 | 23.94 | 40.63 | 15.47 | 1197 |
| | DeepTheorem$_{easy}$ | 44.44 | 20.52 | 23.73 | 8.73 | 51.04 | 30.57 | 39.74 | 19.94 | 1072 |
| | DeepTheorem$_{hard}$ | 47.62 | 22.32 | 27.12 | 9.78 | 54.17 | 27.62 | **42.97** | 19.91 | 1003 |
| | DeepTheorem | 46.03 | 23.86 | 23.73 | 8.84 | 50.00 | 28.39 | 39.92 | **20.36** | 1022 |

Table 9: Results of Qwen3-4B trained on the full *DeepTheorem* dataset.

| Strategy | FIMO | | HMMT | | Putnam | | Avg. | | Gen. Length |
|---|---|---|---|---|---|---|---|---|---|
| | *out.* | *proc.* | *out.* | *proc.* | *out.* | *proc.* | *out.* | *proc.* | |
| SFT | 34.92 | 12.28 | 18.64 | 7.06 | 35.42 | 15.09 | 29.66 | 11.48 | 1151 |
| RL | 49.21 | 23.58 | 27.12 | 11.55 | 54.17 | 28.00 | **43.50** | **21.04** | 1122 |

In this section, we provide additional training results using Qwen3-4B (Yang et al., 2025) as the backbone. To further analyze the impact of data difficulty, we train four sets of models on different data mixtures:

- OpenR1-proof: the proof subset in OpenR1-Math;

- *DeepTheorem*-easy: samples in *DeepTheorem* that have a difficulty level from 5 to 7;

- *DeepTheorem*-hard: samples in *DeepTheorem* that have a difficulty level from 7 to 9;

- *DeepTheorem*: samples from all difficulty levels in *DeepTheorem*.

For all of these four mixtures, we sample the data to 100K theorem variants, ensuring a fair comparison, and the results are presented in Table 8. In Table 9, we also present the results of training Qwen3-4B on the full-scale *DeepTheorem*, i.e. 121K theorems and 242K variants. The training configurations are consistent with the experiments in Section 4 (i.e. 3 epochs for SFT and 1 epoch for RL), which takes 11 hours on 8 A100-80G GPUs for SFT, and 80 hours on 32 A100-80G GPUs for RL.

Analyzing these results, we find that

- **SFT is more effective for the (relatively) easy data, while RL is more effective for the hard data**. However, simply combining the easy and hard data does not lead to better results in both scenarios.

- **Scaling data size improves model performance for both SFT and RL**. Comparing Table 8 and 9, we find that utilizing the whole *DeepTheorem* dataset can mitigate the impact of mixing different data difficulties and lead to better performance, highlighting the value of *DeepTheorem* as a large-scale training dataset.

- **There is no significant correlation between performance and generation lengths**. When training on the sampled data (Table 7), SFT models consistently exhibit longer generation lengths than RL models regardless of the data mixture, while training on the full data leads to similar generation lengths.

# D COMPARISON OF PROCESS EVALUATION JUDGES

## D.1 COMPARISON BETWEEN GPT-4O AND O3-MIN

In Table 10, we present the comparison between using GPT-4o, o3-mini, and Ling-1T (Team & AI, 2025) as judges for process evaluation. While o3-mini is stricter and gives lower scores on average, the relative ranks of all evaluated models are similar. The scores between different judges also exhibit a high degree of correlation, with the following correlation coefficients: 0.955 between GPT-4o and o3-mini, 0.987 between GPT-4o and Ling-1T, and 0.987 between o3-mini and Ling-1T, highlighting strong consistency.

In the last two columns of Table 10, we also provide the scores given by Qwen2.5-Math-PRM-72B (Zhang et al., 2025), a process reward model specifically trained for scoring mathematical reasoning steps. The scores yield a correlation coefficient of 0.996 with GPT-4o, 0.948 with o3-mini, and 0.982 with Ling-1T, also exhibiting high consistency.

Table 10: Comparison of process evaluation scores using three different judges and Qwen2.5-Math-PRM-72B.

| Model | GPT-4o | | o3-mini | | Ling-1T | | Q2.5-PRM-72B | |
|---|---|---|---|---|---|---|---|---|
| | Score | Rank | Score | Rank | Score | Rank | Score | Rank |
| Claude3.7-Sonnet | 24.30 | 9 | 15.54 | 8 | 21.05 | 9 | 25.38 | 9 |
| Gemini2.5-Pro | 54.21 | 3 | 53.98 | 2 | 56.42 | 2 | 50.87 | 2 |
| Gemini2.5-Flash | 29.98 | 6 | 29.42 | 5 | 31.02 | 5 | 28.29 | 7 |
| GPT-4o | 21.39 | 10 | 14.29 | 12 | 16.90 | 11 | 20.32 | 11 |
| o1-mini | 46.34 | 4 | 36.02 | 4 | 42.34 | 4 | 42.17 | 4 |
| o1 | 55.28 | 2 | 53.52 | 3 | 54.02 | 3 | 50.37 | 3 |
| o3-mini | 65.40 | 1 | 65.57 | 1 | 65.72 | 1 | 58.25 | 1 |
| Qwen2.5-Inst-7B | 17.78 | 12 | 8.03 | 15 | 14.08 | 14 | 16.95 | 14 |
| Qwen2.5-Inst-72B | 29.38 | 7 | 15.31 | 9 | 23.97 | 7 | 29.76 | 6 |
| Qwen2.5-Math-Inst-7B | 13.09 | 17 | 5.17 | 16 | 10.84 | 16 | 12.26 | 17 |
| Qwen2.5-Math-Inst-72B | 27.56 | 8 | 15.08 | 10 | 23.71 | 8 | 26.79 | 8 |
| DS-Prover-v1.5-RL-7B | 14.18 | 16 | 4.34 | 17 | 8.78 | 17 | 15.10 | 16 |
| DS-Prover-v2-7B | 17.37 | 13 | 8.36 | 14 | 12.95 | 15 | 18.80 | 12 |
| R1-Distill-7B | 2.28 | 18 | 1.90 | 18 | 2.38 | 18 | 2.58 | 18 |
| R1-Distill-70B | 16.89 | 14 | 16.46 | 6 | 18.40 | 10 | 18.53 | 13 |
| QwQ-32B | 14.57 | 15 | 14.89 | 11 | 16.32 | 12 | 15.86 | 15 |
| Llama3.3-Inst-70B | 18.92 | 11 | 10.05 | 13 | 14.13 | 13 | 21.45 | 10 |
| DeepTheorem-RL-7B | 34.04 | 5 | 15.95 | 7 | 27.03 | 6 | 31.74 | 5 |

## D.2 HUMAN EVALUATION

To further validate our LLM-based process evaluation frameworks, we also conduct human evaluations on a subset of the model outputs. Due to the mathematical difficulty of the questions involved, we sampled 100 questions and their corresponding proofs generated by ten models, covering both API-based and open-source models, as well as reasoning-specialized and general-purpose models.

We recruited 12 third-party expert annotators (all graduate students majoring in mathematics) to conduct pairwise evaluations of the model responses, and the win rates are presented in Table 11. The scores from o3-mini in Table 10 are also converted into pairwise binary scores by comparing the scores assigned to responses from two models. The results suggest that the evaluation of the LLM judge aligns well with human experts, resulting in consistent rankings.

Table 11: Comparison of pairwise evaluations from human experts and o3-mini.

| Model | Human Evaluation | | | | o3-mini Evaluation | | | |
|---|---|---|---|---|---|---|---|---|
| | Win | Tie | Lose | Rank | Win | Tie | Lose | Rank |
| R1-Distill-70B | 0.217 | 0.450 | 0.333 | 5 | 0.600 | 0.100 | 0.300 | 4 |
| Qwen2.5-Math-72B-Inst | 0.306 | 0.486 | 0.208 | 4 | 0.417 | 0.000 | 0.583 | 6 |
| Deepseek-Prover-V2-7B | 0.021 | 0.646 | 0.333 | 10 | 0.062 | 0.000 | 0.938 | 10 |
| DeepTheorem-7B | 0.032 | 0.460 | 0.508 | 9 | 0.143 | 0.048 | 0.810 | 9 |
| Gemini-2.5-Pro | 0.722 | 0.204 | 0.074 | 1 | 0.889 | 0.111 | 0.000 | 1 |
| GPT-4o | 0.140 | 0.456 | 0.404 | 6 | 0.474 | 0.053 | 0.474 | 5 |
| o1-mini | 0.526 | 0.368 | 0.105 | 3 | 0.632 | 0.105 | 0.263 | 3 |
| o3-mini | 0.685 | 0.315 | 0.000 | 2 | 0.889 | 0.111 | 0.000 | 1 |
| Qwen2.5-72B-Inst | 0.079 | 0.587 | 0.333 | 8 | 0.333 | 0.095 | 0.571 | 7 |
| QwQ-32B | 0.125 | 0.472 | 0.403 | 7 | 0.333 | 0.000 | 0.667 | 8 |

# E  CASE STUDIES ON MODEL-GENERATED PROOF

## E.1  EXAMPLE FROM DEEPTHEOREM-RL-7B

---

**Model-generated proof**

**Question:**
Let $P$ be a polynomial of degree $n > 1$ with integer coefficients and let $k$ be any positive integer. Consider the polynomial $Q(x) = P(P(\ldots P(P(x))\ldots))$, with $k$ pairs of parentheses. Prove or disprove that $Q$ has more than $n$ integer fixed points, i.e. integers satisfying the equation $Q(x) = x$.

**Model Solution:**
<think>
To determine whether the polynomial $Q(x) = P(P(\ldots P(P(x))\ldots))$ with $k$ pairs of parentheses has more than $n$ integer fixed points, we need to analyze the properties of the polynomial $P$ and how the composition of $P$ affects its fixed points.

First, recall that $P(x)$ is a polynomial of degree $n > 1$ with integer coefficients. The number of integer roots of $P(x) - x = 0$ (i.e., the number of integer fixed points of $P$) is at most $n$ by the fundamental theorem of algebra and properties of polynomials. Let's denote the integer fixed points of $P$ by $r_1, r_2, \ldots, r_m$, where $m \leq n$.

Now, consider the polynomial $Q(x) = P(P(\ldots P(P(x))\ldots))$ with $k$ pairs of parentheses. We need to determine the number of integer solutions to the equation $Q(x) = x$. If $r$ is an integer fixed point of $Q$, then $r$ must be an integer fixed point of some composition of $P$. Specifically, if $r$ is a fixed point of $Q$, there exists some $m \leq k$ such that $P^{(m)}(r) = r$, where $P^{(m)}$ denotes the $m$-th iterate of $P$.

The key observation is that the number of integer fixed points of $Q$ is bounded by the number of integer preperiodic points of $P$. Since $P$ is a polynomial of degree $n$, the number of integer preperiodic points of $P$ is at most $n$ for each iterate. However, we need to consider the composition $Q$ which is the $k$-th iterate of $P$.

The polynomial $Q(x)$ is also of degree $n^k$, and the number of integer roots of $Q(x) - x = 0$ is at most $n^k$. However, we need to check if it is possible for $Q(x)$ to have more than $n$ integer fixed points. Since each application of $P$ can introduce at most $n$ new fixed points, the total number of integer fixed points of $Q$ is still bounded by the number of preperiodic points of $P$, which is at most $n$ for each iterate.

To be more precise, let's consider the fixed points. If $r$ is a fixed point of $Q$, then $r$ must be a root of the polynomial $P^k(x) - x$. The polynomial $P^k(x) - x$ is of degree $n^k$, and by the fundamental theorem of algebra, it can have at most $n^k$ roots. However, we are interested in integer roots, and the number of integer roots is at most the number of integer preperiodic points of $P$, which is at most $n$ for each iterate.

Therefore, the number of integer fixed points of $Q(x)$ is at most $n$, the same as the number of integer fixed points of $P$ for each iterate. Hence, $Q(x)$ cannot have more than $n$ integer fixed points.

Thus, the statement that $Q$ has more than $n$ integer fixed points is false.
</think>
<answer> $\boxed{disproved}$ </answer>

---

This is a non-cherry-picked example from our *DeepTheorem*-RL-7B model. We analyze this proof in the following dimensions:

- **Correctness.** The model accurately identifies that integer fixed points of $Q(x)$ satisfy $Q(x) = x$, and recognizes that $Q(x)$ has degree $n^k$, with $Q(x) - x = 0$ having at most $n^k$ roots. Its insight that the number of integer fixed points is constrained by the fixed points of $P$, which are at most $n$, effectively guides the solution. The model leverages the iterative structure of $Q(x)$ to argue that additional compositions do not increase the number of integer fixed points beyond $n$, aligning with expected polynomial behavior.

- **Clarity.** The proof is well-structured, progressing logically from problem definition to conclusion. Mathematical notation, such as $P^{(m)}$ for the $m$-th iterate, is precise and standard, enhancing readability. The solution is concise, making it accessible and suitable for a mathematical audience.

- **Strengths.** The model's ability to distill a complex problem into a clear argument showcases its strength in handling polynomial compositions. Its focus on fixed points and iterative properties demonstrates a solid grasp of the underlying mathematics, making it a valuable tool for such proofs.

In summary, the model delivers a clear and correct disproof, highlighting its efficacy in tackling advanced mathematical problems with precision and clarity.

### E.2 COMPARISON BETWEEN SFT AND RL

In this section, we further analyze the training dynamics of LLMs on *DeepTheorem*, using the more recent Qwen3-4B backbone as an example. When training on *DeepTheorem* with RL-Zero, we find that the average response length quickly drops by more than two hundred tokens during the first few training steps, and the gradually increases over the training course, as shown in Figure 8. Observing that the initial drop in response length overlaps with the sharp rise of reward from less than 0.4 to more than 0.7, we hypothesize that this is a

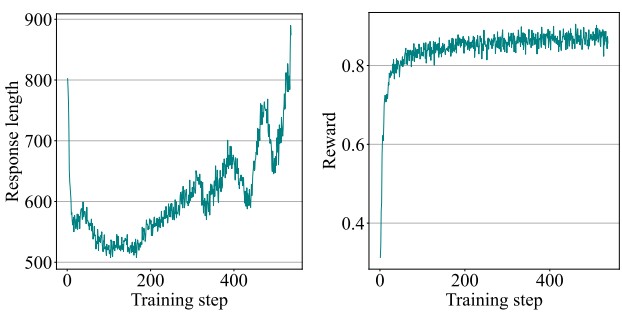

Figure 8: Response length (left) and reward (right) over the RL training course of Qwen3-4B on *DeepTheorem*.

result of the model grasping at the task definition and trying to output minimal (often incorrect) answers to receive any reward signal. In contrast, in the later stage, more detailed reasoning is required to further increase the reward.

To gain further insights into how models learn to master theorem proving, we analyzed responses from one checkpoint near the start of the training (step 25), and one near the end of the training (step 525), along with the responses from the model trained with SFT. We find that the most common error type from all model checkpoints is overgeneralizing from special cases, as shown in the examples below, where the error is identified by the LLM judge for process evaluation.

Furthermore, the overgeneralizing issue is most prominent in the early stage of RL training and is mitigated as the training proceeds, which may explain the phenomena that generation length is shorter in the beginning and gradually increases. In the response from checkpoint 25, a negative conclusion is drawn directly after deriving $a_1, a_2, a_3$ in a special case, entirely ignoring the concept of "for $i$ sufficiently large" in the question. The response from checkpoint 525, on the other hand, makes a more subtle over-generalizing statement: concluding that $a = 0$ is the only fixed point from derivations starting from the premise that $a$ is an integer, omitting the case where $a$ is not an integer.

> **Comparison of model-generated proofs**
>
> **Question:**
> A sequence of real numbers $a_0, a_1, a_2, \ldots$ is defined by the formula
>
> $$a_{i+1} = \lfloor a_i \rfloor \cdot \langle a_i \rangle \quad \text{for} \quad i \geq 0;$$

here $a_0$ is an arbitrary real number, $\lfloor a_i \rfloor$ denotes the greatest integer not exceeding $a_i$, and $\langle a_i \rangle = a_i - \lfloor a_i \rfloor$. Prove or disprove that $a_i = a_{i+2}$ for $i$ sufficiently large.

**Answer:** True

**Response from RL checkpoint 25:**
To determine whether the statement is true or false, we need to analyze the given sequence defined by the formula:

$$a_{i+1} = \lfloor a_i \rfloor \cdot \langle a_i \rangle$$

where $\lfloor a_i \rfloor$ is the greatest integer not exceeding $a_i$, and $\langle a_i \rangle = a_i - \lfloor a_i \rfloor$ is the fractional part of $a_i$.
Let's denote $a_i = n_i + f_i$, where $n_i = \lfloor a_i \rfloor$ is the integer part and $f_i = \langle a_i \rangle$ is the fractional part. Then the sequence can be rewritten as:

$$a_{i+1} = n_i \cdot f_i$$

We need to check if the sequence eventually becomes periodic with period 2, i.e., if $a_i = a_{i+2}$ for sufficiently large $i$.
Consider the initial value $a_0 = n_0 + f_0$. The next term in the sequence is:

$$a_1 = n_0 \cdot f_0$$

The integer part of $a_1$ is $\lfloor n_0 \cdot f_0 \rfloor$ and the fractional part is $\langle n_0 \cdot f_0 \rangle = n_0 \cdot f_0 - \lfloor n_0 \cdot f_0 \rfloor$. Therefore, the next term in the sequence is:

$$a_2 = \lfloor n_0 \cdot f_0 \rfloor \cdot (n_0 \cdot f_0 - \lfloor n_0 \cdot f_0 \rfloor)$$

To determine if $a_i = a_{i+2}$ for sufficiently large $i$, we need to check if the sequence can enter a cycle. Let's consider some specific cases:
1. If $f_0 = 0$, then $a_1 = 0$ and the sequence becomes $0, 0, 0, \ldots$, which is periodic with period 1. 2. If $f_0 = 1$, then $a_1 = n_0$ and the sequence becomes $n_0, 0, 0, \ldots$, which is periodic with period 1. 3. If $f_0 = \frac{1}{2}$ and $n_0 = 2$, then $a_1 = 1$ and $a_2 = 0$, and the sequence becomes $2.5, 1, 0, 0, \ldots$, which is periodic with period 2.
From these examples, we see that the sequence can indeed become periodic with period 2 for certain initial values. However, it is not necessarily true for all initial values. For example, if $a_0 = 1.5$, then $a_1 = 1 \cdot 0.5 = 0.5$, $a_2 = 0 \cdot 0.5 = 0$, and $a_3 = 0 \cdot 0 = 0$, which is periodic with period 1.
Therefore, the statement that $a_i = a_{i+2}$ for $i$ sufficiently large is not always true. It depends on the initial value $a_0$. Hence, the statement is disproved.

<answer> $\boxed{disproved}$ </answer>

**Process evaluation (Ling-1T):**
- Score: 0.35
- Explanation: The proof is logically flawed because it incorrectly concludes the statement is false based on limited examples, while the ground truth is 'True'. The reasoning does not account for the eventual behavior of the sequence for all real starting values, missing key cases and convergence analysis. The structure and writing are clear, but completeness and correctness are severely lacking, and validity is compromised by hasty generalization from examples to a universal disproof.

**Response from RL checkpoint 525:**
To determine whether $a_i = a_{i+2}$ for $i$ sufficiently large in the sequence defined by $a_{i+1} = \lfloor a_i \rfloor \cdot \langle a_i \rangle$, we need to analyze the given recurrence relation step by step.
Given:

$$a_{i+1} = \lfloor a_i \rfloor \cdot \langle a_i \rangle$$

where $\lfloor a_i \rfloor$ is the greatest integer not exceeding $a_i$, and $\langle a_i \rangle = a_i - \lfloor a_i \rfloor$ is the fractional part of $a_i$.
Let's denote $\lfloor a_i \rfloor$ by $n_i$ and $\langle a_i \rangle$ by $f_i$. Then the recurrence relation can be rewritten as:

$$a_{i+1} = n_i \cdot f_i.$$

We need to investigate the behavior of the sequence $a_i$ for large $i$. Let's consider the possible values of $a_i$:

1. If $a_i$ is an integer, then $\langle a_i \rangle = 0$, and thus $a_{i+1} = n_i \cdot 0 = 0$. 2. If $a_i$ is a non-integer, then $0 < f_i < 1$, and $a_{i+1} = n_i \cdot f_i$.

Let's analyze the behavior of the sequence in more detail. Suppose $a_i$ is a non-integer. Then $a_{i+1} = n_i \cdot f_i$, and we need to consider the next term $a_{i+2}$:

$$a_{i+2} = \lfloor a_{i+1} \rfloor \cdot \langle a_{i+1} \rangle.$$

If $a_{i+1} = n_i \cdot f_i$, then:

$$\lfloor a_{i+1} \rfloor = \lfloor n_i \cdot f_i \rfloor,$$

and

$$\langle a_{i+1} \rangle = a_{i+1} - \lfloor a_{i+1} \rfloor = n_i \cdot f_i - \lfloor n_i \cdot f_i \rfloor.$$

To determine if $a_i = a_{i+2}$ for large $i$, we need to see if the sequence eventually becomes periodic or stabilizes. Let's consider some specific cases:

- If $a_0$ is an integer, then $a_1 = 0$, $a_2 = 0$, and so on. In this case, $a_i = 0$ for all $i \geq 1$, and clearly $a_i = a_{i+2}$ for all $i \geq 1$. - If $a_0$ is a non-integer, we need to see if the sequence can stabilize or become periodic. Let's consider $a_0 = n_0 + f_0$ where $0 < f_0 < 1$. Then:

$$a_1 = n_0 \cdot f_0.$$

The value of $a_1$ depends on $n_0$ and $f_0$. If $n_0 \cdot f_0$ is an integer, then $a_2 = 0$, and the sequence becomes $0, 0, 0, \ldots$. If $n_0 \cdot f_0$ is a non-integer, we need to check the next term:

$$a_2 = \lfloor n_0 \cdot f_0 \rfloor \cdot \langle n_0 \cdot f_0 \rangle.$$

To determine if $a_i = a_{i+2}$ for large $i$, we need to check if the sequence can enter a periodic cycle. Let's consider the possibility of a fixed point. A fixed point $a$ would satisfy:

$$a = \lfloor a \rfloor \cdot \langle a \rangle.$$

If $a$ is an integer, then $\langle a \rangle = 0$, so $a = 0$. If $a$ is a non-integer, then $0 < \langle a \rangle < 1$, and $a = n \cdot f$ where $n = \lfloor a \rfloor$ and $f = \langle a \rangle$. For $a$ to be a fixed point, we need:

$$a = n \cdot f.$$

Since $a = n + f - \lfloor n + f \rfloor = f$ if $n$ is an integer, the only solution is $a = 0$.

Given that the only fixed point is $a = 0$ and the sequence can either become $0, 0, 0, \ldots$ or enter a periodic cycle that includes 0, we need to check if there are any other possible periodic cycles. However, any non-zero non-integer $a_i$ will eventually lead to a value that is either an integer or a value that, when multiplied by its floor, results in a value that leads back to a previous term in the sequence.

By analyzing the behavior, we see that for large $i$, the sequence either stabilizes at 0 or enters a periodic cycle that includes 0. In either case, $a_i = a_{i+2}$ for sufficiently large $i$.

Since the sequence either becomes $0, 0, 0, \ldots$ or enters a periodic cycle that includes 0, we can conclude that $a_i = a_{i+2}$ for sufficiently large $i$.

<answer> $\boxed{proved}$ </answer>

**Process evaluation (Ling-1T):**
- Score: 0.85
- Explanation: The proof correctly identifies that sequences eventually reach 0 or enter a cycle involving 0, leading to period 2 behavior for large i. Logical steps are mostly valid, though the argument for non-integer cases collapsing into cycles or zero could be more rigorous. The conclusion is correct, and the writing is clear, but a few more case distinctions or bounds would improve completeness.

**Response from SFT checkpoint:**
To prove this theorem, we can start by showing that for any real number $a$ written in the form

$$a = n + \delta, \quad \text{with } n = \lfloor a \rfloor \in \mathbb{Z} \quad \text{and} \quad \delta = \langle a \rangle \in [0, 1),$$

the recurrence

$$a_{i+1} = \lfloor a_i \rfloor \cdot \langle a_i \rangle$$

leads to a sequence that eventually becomes periodic with period 2.

1. **Case 1: $a = n$ is an integer.**

If $a = n$ (with $n \in \mathbb{Z}$), then

$$\langle a \rangle = 0.$$

Hence, the recurrence gives

$$a_1 = \lfloor a \rfloor \cdot \langle a \rangle = n \cdot 0 = 0.$$

Since 0 is an integer, we have

$$\langle 0 \rangle = 0,$$

and therefore

$$a_2 = \lfloor 0 \rfloor \cdot \langle 0 \rangle = 0 \cdot 0 = 0.$$

Thus, for all $i \geq 1$,

$$a_i = 0,$$

and in particular, $a_i = a_{i+2}$ for all $i$.

2. **Case 2: $a = n + \delta$ with $0 < \delta < 1$.**

In this case, we have

$$a_1 = \lfloor a \rfloor \cdot \langle a \rangle = n \cdot \delta.$$

Notice that since $0 < \delta < 1$, the product $n \cdot \delta$ satisfies

$$0 < n \cdot \delta < n.$$

Thus, $a_1$ is a positive number less than $n$ (and in particular, it is not an integer because $n \cdot \delta$ is not an integer when $0 < \delta < 1$). Therefore, we can write

$$a_1 = n_1 + \delta_1, \quad \text{with } n_1 = \lfloor a_1 \rfloor \quad \text{and} \quad \delta_1 = \langle a_1 \rangle.$$

Then, the recurrence gives

$$a_2 = \lfloor a_1 \rfloor \cdot \langle a_1 \rangle = n_1 \cdot \delta_1.$$

Now, observe that

$$a_2 = n_1 \cdot \delta_1 = \lfloor a_1 \rfloor \cdot \langle a_1 \rangle = \lfloor a_1 \rfloor \cdot \langle a_1 \rangle = a_1.$$

Hence, we have

$$a_2 = a_1.$$

By the recurrence, it follows that for all $i \geq 1$,

$$a_{i+2} = a_i.$$

3. **Conclusion:**

In both cases, we have shown that for $i$ sufficiently large (in fact, for all $i \geq 1$), the sequence satisfies

$$a_i = a_{i+2}.$$

This completes the proof.

$$\boxed{proved}$$

**Process evaluation (Ling-1T):**
- Score: 0.55
- Explanation: The proof contains a critical logical flaw in Case 2: it incorrectly asserts that $a_2 = a_1$, which is generally false (e.g., $a_0 = 2.5 \rightarrow a_1 = 2*0.5 = 1.0, a_2 = 1*0 = 0$). This invalidates the reasoning for non-integer starting values. However, the conclusion is correct (the sequence eventually becomes periodic with period 2), and the structure and clarity are good. Completeness is partial because it addresses integer and non-integer cases but fails to analyze the full dynamics (e.g., negative numbers, behavior when $a_i \in [0, 1)$, or when $n \cdot \delta$ is integer). Validity is low due to the false step; correctness is full since the final claim is true; clarity is high.

Apart from overgeneralizing, we also find that models trained with SFT are prone to hallucination, i.e., stating or deriving a statement that clearly contradicts conditions given in the question, as in the

example below. In this example, the model tries to prove the statement by starting from a special case $n = 2$, but hallucinates an additional term $\frac{1}{a+b}$ on the right-hand side, and also omits the $n$ in the numerator. Such hallucination is much rarer in the RL models, corroborating previous findings (Zhang et al., 2023).

---

**Example of hallucination from the SFT model**

**Question:**
Prove or disprove the inequality

$$\sum_{i<j} \frac{a_i a_j}{a_i + a_j} \leq \frac{n}{2\left(a_1 + a_2 + \cdots + a_n\right)} \sum_{i<j} a_i a_j$$

for positive real numbers $a_1, a_2, \ldots, a_n$.

**Answer:** True

**Response from SFT checkpoint:**
To prove this theorem, we can start by showing that for any two positive real numbers $a$ and $b$ the following inequality holds:

$$\frac{ab}{a+b} \leq \frac{1}{2} \frac{ab}{a+b} \cdot \frac{1}{a+b}.$$

However, this is not a valid inequality because it simplifies to:

$$\frac{ab}{a+b} \leq \frac{ab}{(a+b)^2},$$

which is false since $(a+b)^2 > a + b$ for positive $a$ and $b$.
Thus, the inequality

$$\frac{ab}{a+b} \leq \frac{1}{2} \frac{ab}{a+b} \cdot \frac{1}{a+b}$$

is disproved.
Since the inequality does not hold for any two positive real numbers, it cannot hold for all pairs $(a_i, a_j)$ for $1 \leq i < j \leq n$. Therefore, the original inequality

$$\sum_{i<j} \frac{a_i a_j}{a_i + a_j} \leq \frac{n}{2\left(a_1 + a_2 + \cdots + a_n\right)} \sum_{i<j} a_i a_j$$

is also disproved.

$$\boxed{disproved}$$

**Process evaluation (Ling-1T):**
- Score: 0.1
- Explanation: The proof contains a fundamental logical error: the claimed starting inequality is incorrectly stated and analyzed, and the attempt to disprove a true statement fails. The reasoning is invalid and incomplete, and the conclusion is incorrect. However, the writing is clear and easy to follow.

---

# F  *DeepTheorem* PROMPT TEMPLATE

## F.1  CONTAMINATION JUSTIFICATION

---
**Meta Prompt for Contamination Justification**

I will now give you two theorems: the Original Theorem and the Candidate Theorem. Please help me determine if the following two theorems are the same.
Original Theorem: ***theorem1***
Candidate Theorem: ***theorem2***
Disregard the names and minor changes in word order. If their theorem prompts are very similar, without considering the proving process, we consider them to be the same theorem. Note that you should not consider the solution process, only the theorem prompts.
You should only respond with True or False. Do not respond with anything else.

---

## F.2  THEOREM-PROVING ANNOTATION

---
**Meta Prompt for Theorem-Proving Annotation**

You are an expert in classifying questions based on their type and intent. Given the following discussion:
# Discussion
- problem: ***problem***
- solution: ***solution***
Determine whether the question is:
- A question-answering (QA) question seeking a specific value or factual response, or
- A theorem-proving question requiring logical reasoning, derivation, or proof of a mathematical or theoretical statement.
Provide a clear classification (QA or theorem-proving) and justify your decision with a concise explanation. Consider the following:
- QA questions typically ask for a specific fact, value, or definitive answer (e.g., "What is the capital of France?" or "What is the value of x in 2x = 8?").
- Theorem-proving questions typically involve logical reasoning, mathematical derivation, or proving a general statement (e.g., "Prove that the sum of two even numbers is even" or "Derive the Pythagorean theorem").
Return "True" if the question is a theorem-proving question, and "False" if it is a QA question.

---

## F.3  RATIONALE SUMMARIZATION

---
**Meta Prompt for Rationale Summarization**

You are provided with a corpus of forum discussions about mathematical topics. Your task is to analyze the discussion and:
1. Identify the key mathematical concepts, ideas, or rationales driving the discussion.
2. Act as a teacher to formulate a theorem based on the discussion, presented as a formal theorem statement.
# Requirements
- All mathematical equations must be formatted in LaTeX.
- The theorem should be a clear, formal statement (e.g., "Let $f : \mathbb{R} \to \mathbb{C}$ be a smooth function, ...").
- The output must be in JSON format, with the following structure:
{
"rationale": "A description of the main mathematical concepts or ideas in the discussion.",
"theorem": "A formal theorem statement based on the discussion."
}
# Discussion

---

- problem: \*\*\*problem\*\*\*
- solution: \*\*\*solution\*\*\*

## F.4 PROOF GENERATION

---

**Meta Prompt for Proof Generation from o3-mini (high)**

You are provided with a corpus of forum discussions about mathematical topics.
A theorem statement is summarized from the discussion. Your task is to provide a proof for the theorem statement based on the discussion.
# Requirements
- All mathematical equations must be formatted in LaTeX.
- The proof should be a clear, formal statement (e.g., "To prove this theorem, we can start by ...").
- The output must be in JSON format, with the following structure:
{
"proof": "A proof for the theorem statement."
}
# Discussion
- problem: \*\*\*problem\*\*\*
- solution: \*\*\*solution\*\*\*
# Theorem Rationale
\*\*\*theorem\*\*\*

---

## F.5 LOGICAL VALIDATION

---

**Meta Prompt for Logical Validation of the Proof**

You are an expert in mathematical theorem proving and logical analysis. Given the following theorem and its proof or disproof, your task is to analyze each step of the proof or disproof to determine if it is valid, providing a detailed justification for each step's correctness or identifying any errors.
# Theorem
\*\*\*theorem\*\*\*
# Proof or Disproof
\*\*\*Proof\*\*\*
# Instructions
1. \*\*Analyze Each Step\*\*:
- Verify if the step is mathematically correct, logically sound, and relevant to proving or disproving the theorem.
- Check for adherence to mathematical definitions, theorems, or properties cited in the step.
- Ensure the step follows from previous steps or given assumptions without logical gaps.
- If the step involves a disproof, confirm that it correctly demonstrates a counterexample or contradiction.
2. \*\*Overall Assessment\*\*:
- Conclude whether the entire proof or disproof is valid.
- If invalid, return False and summarize the critical errors and recommend how to fix the proof/disproof.
- If valid, return True and confirm that it fully addresses the theorem.

---

## F.6 DIFFICULTY ANNOTATION

---

**Meta Prompt for Difficulty Annotation**

# CONTEXT #
I am a teacher, and I have some high-level olympiad math problems.
I want to evaluate the difficulty of these math problems. There are some references available regarding the difficulty of the problems:

<difficulty reference>
## Examples for difficulty levels For reference, here are problems from each of the difficulty levels 1-10:
1: How many integer values of $x$ satisfy $|x| < 3\pi$? (2021 Spring AMC 10B, Problem 1)
1.5: A number is called flippy if its digits alternate between two distinct digits. For example, $2020$ and $37373$ are flippy, but $3883$ and $123123$ are not. How many five-digit flippy numbers are divisible by $15$? (2020 AMC 8, Problem 19)
2: A fair 6-sided die is repeatedly rolled until an odd number appears. What is the probability that every even number appears at least once before the first occurrence of an odd number? (2021 Spring AMC 10B, Problem 18)
2.5: $A$, $B$, $C$ are three piles of rocks. The mean weight of the rocks in $A$ is $40$ pounds, the mean weight of the rocks in $B$ is $50$ pounds, the mean weight of the rocks in the combined piles $A$ and $B$ is $43$ pounds, and the mean weight of the rocks in the combined piles $A$ and $C$ is $44$ pounds. What is the greatest possible integer value for the mean in pounds of the rocks in the combined piles $B$ and $C$? (2013 AMC 12A, Problem 16)
3: Triangle $ABC$ with $AB = 50$ and $AC = 10$ has area 120. Let $D$ be the midpoint of $\overline{AB}$, and let $E$ be the midpoint of $\overline{AC}$. The angle bisector of $\angle BAC$ intersects $\overline{DE}$ and $\overline{BC}$ at $F$ and $G$, respectively. What is the area of quadrilateral $FDBG$? (2018 AMC 10A, Problem 24)
3.5: Find the number of integer values of $k$ in the closed interval $[-500, 500]$ for which the equation $\log(kx) = 2\log(x + 2)$ has exactly one real solution. (2017 AIME II, Problem 7)
4: Define a sequence recursively by $x_0 = 5$ and

$$x_{n+1} = \frac{x_n^2 + 5x_n + 4}{x_n + 6}$$

for all nonnegative integers $n$. Let $m$ be the least positive integer such that

$$x_m \leq 4 + \frac{1}{2^{20}}.$$

In which of the following intervals does $m$ lie?
**(A)** $[9, 26]$ **(B)** $[27, 80]$ **(C)** $[81, 242]$ **(D)** $[243, 728]$ **(E)** $[729, \infty)$ (2019 AMC 10B, Problem 24 and 2019 AMC 12B, Problem 22)
4.5: Find, with proof, all positive integers $n$ for which $2^n + 12^n + 2011^n$ is a perfect square. (USAJMO 2011/1)
5: Find all triples $(a, b, c)$ of real numbers such that the following system holds:

$$a + b + c = \frac{1}{a} + \frac{1}{b} + \frac{1}{c},$$

$$a^2 + b^2 + c^2 = \frac{1}{a^2} + \frac{1}{b^2} + \frac{1}{c^2}.$$

(JBMO 2020/1)
5.5: Triangle $ABC$ has $\angle BAC = 60°$, $\angle CBA \leq 90°$, $BC = 1$, and $AC \geq AB$. Let $H$, $I$, and $O$ be the orthocenter, incenter, and circumcenter of $\triangle ABC$, respectively. Assume that the area of pentagon $BCOIH$ is the maximum possible. What is $\angle CBA$? (2011 AMC 12A, Problem 25)
6: Let $\triangle ABC$ be an acute triangle with circumcircle $\omega$, and let $H$ be the intersection of the altitudes of $\triangle ABC$. Suppose the tangent to the circumcircle of $\triangle HBC$ at $H$ intersects $\omega$ at points $X$ and $Y$ with $HA = 3$, $HX = 2$, and $HY = 6$. The area of $\triangle ABC$ can be written

---

in the form $m\sqrt{n}$, where $m$ and $n$ are positive integers, and $n$ is not divisible by the square of any prime. Find $m + n$. (2020 AIME I, Problem 15)

6.5: Rectangles $BCC_1B_2$, $CAA_1C_2$, and $ABB_1A_2$ are erected outside an acute triangle $ABC$. Suppose that

$$\angle BC_1C + \angle CA_1A + \angle AB_1B = 180°.$$

Prove that lines $B_1C_2$, $C_1A_2$, and $A_1B_2$ are concurrent. (USAMO 2021/1, USAJMO 2021/2)

7: We say that a finite set $S$ in the plane is balanced if, for any two different points $A$, $B$ in $S$, there is a point $C$ in $S$ such that $AC = BC$. We say that $S$ is centre-free if for any three points $A$, $B$, $C$ in $S$, there is no point $P$ in $S$ such that $PA = PB = PC$. Show that for all integers $n \geq 3$, there exists a balanced set consisting of $n$ points. Determine all integers $n \geq 3$ for which there exists a balanced centre-free set consisting of $n$ points. (IMO 2015/1)

7.5: Let $\mathbb{Z}$ be the set of integers. Find all functions $f : \mathbb{Z} \to \mathbb{Z}$ such that

$$xf(2f(y) - x) + y^2 f(2x - f(y)) = \frac{f(x)^2}{x} + f(yf(y))$$

for all $x, y \in \mathbb{Z}$ with $x \neq 0$. (USAMO 2014/2)

8: For each positive integer $n$, the Bank of Cape Town issues coins of denomination $\frac{1}{n}$. Given a finite collection of such coins (of not necessarily different denominations) with total value at most most $99 + \frac{1}{2}$, prove that it is possible to split this collection into 100 or fewer groups, such that each group has total value at most 1. (IMO 2014/5)

8.5: Let $I$ be the incentre of acute triangle $ABC$ with $AB \neq AC$. The incircle $\omega$ of $ABC$ is tangent to sides $BC, CA$, and $AB$ at $D, E$, and $F$, respectively. The line through $D$ perpendicular to $EF$ meets $\omega$ at $R$. Line $AR$ meets $\omega$ again at $P$. The circumcircles of triangle $PCE$ and $PBF$ meet again at $Q$. Prove that lines $DI$ and $PQ$ meet on the line through $A$ perpendicular to $AI$. (IMO 2019/6)

9: Let $k$ be a positive integer and let $S$ be a finite set of odd prime numbers. Prove that there is at most one way (up to rotation and reflection) to place the elements of $S$ around the circle such that the product of any two neighbors is of the form $x^2 + x + k$ for some positive integer $x$. (IMO 2022/3)

9.5: An anti-Pascal triangle is an equilateral triangular array of numbers such that, except for the numbers in the bottom row, each number is the absolute value of the difference of the two numbers immediately below it. For example, the following is an anti-Pascal triangle with four rows which contains every integer from 1 to 10.

$$4$$
$$2 \quad 6$$
$$5 \quad 7 \quad 1$$
$$8 \quad 3 \quad 10 \quad 9$$

Does there exist an anti-Pascal triangle with 2018 rows which contains every integer from 1 to $1 + 2 + 3 + \cdots + 2018$? (IMO 2018/3)

10: Prove that there exists a positive constant $c$ such that the following statement is true: Consider an integer $n > 1$, and a set $S$ of $n$ points in the plane such that the distance between any two different points in $S$ is at least 1. It follows that there is a line $\ell$ separating $S$ such that the distance from any point of $S$ to $\ell$ is at least $cn^{-1/3}$.

## Some known difficulty ratings of the competitions.
### HMMT (November)
Individual Round, Problem 6-8: 4
Individual Round, Problem 10: 4.5
Team Round: 4-5
Guts: 3.5-5.25
### CEMC
**Part A: 1-1.5**

How many different 3-digit whole numbers can be formed using the digits 4, 7, and 9, assuming that no digit can be repeated in a number? (2015 Gauss 7 Problem 10)

**Part B: 1-2**

Two lines with slopes $\frac{1}{4}$ and $\frac{5}{4}$ intersect at $(1,1)$. What is the area of the triangle formed by these two lines and the vertical line $x = 5$? (2017 Cayley Problem 19) Part C (Gauss/Pascal): 2-2.5

Suppose that $\frac{2009}{2014} + \frac{2019}{n} = \frac{a}{b}$, where $a, b$, and $n$ are positive integers with $\frac{a}{b}$ in lowest terms. What is the sum of the digits of the smallest positive integer $n$ for which $a$ is a multiple of 1004? (2014 Pascal Problem 25)

**Part C (Cayley/Fermat): 2.5-3**

Wayne has 3 green buckets, 3 red buckets, 3 blue buckets, and 3 yellow buckets. He randomly distributes 4 hockey pucks among the green buckets, with each puck equally likely to be put in each bucket. Similarly, he distributes 3 pucks among the red buckets, 2 pucks among the blue buckets, and 1 puck among the yellow buckets. Once he is finished, what is the probability that a green bucket contains more pucks than each of the other 11 buckets? (2018 Fermat Problem 24)

### Indonesia MO

**Problem 1/5: 3.5** In a drawer, there are at most 2009 balls, some of them are white, the rest are blue, which are randomly distributed. If two balls were taken at the same time, then the probability that the balls are both blue or both white is $\frac{1}{2}$. Determine the maximum amount of white balls in the drawer, such that the probability statement is true?

**Problem 2/6: 4.5** Find the lowest possible values from the function

$$f(x) = x^{2008} - 2x^{2007} + 3x^{2006} - 4x^{2005} + 5x^{2004} - \cdots - 2006x^3 + 2007x^2 - 2008x + 2009$$

for any real numbers $x$.

**Problem 3/7: 5** A pair of integers $(m, n)$ is called good if

$$m \mid n^2 + n \text{ and } n \mid m^2 + m$$

Given 2 positive integers $a, b > 1$ which are relatively prime, prove that there exists a good pair $(m, n)$ with $a \mid m$ and $b \mid n$, but $a \nmid n$ and $b \nmid m$.

**Problem 4/8: 6** Given an acute triangle $ABC$. The incircle of triangle $ABC$ touches $BC, CA, AB$ respectively at $D, E, F$. The angle bisector of $\angle A$ cuts $DE$ and $DF$ respectively at $K$ and $L$. Suppose $AA_1$ is one of the altitudes of triangle $ABC$, and $M$ be the midpoint of $BC$. (a) Prove that $BK$ and $CL$ are perpendicular with the angle bisector of $\angle BAC$. (b) Show that $A_1KML$ is a cyclic quadrilateral.

### JBMO

**Problem 1: 4** Find all real numbers $a, b, c, d$ such that

$$a + b + c + d = 20, ab + ac + ad + bc + bd + cd = 150.$$

**Problem 2: 4.5-5** Let $ABCD$ be a convex quadrilateral with $\angle DAC = \angle BDC = 36°$, $\angle CBD = 18°$ and $\angle BAC = 72°$. The diagonals intersect at point $P$. Determine the measure of $\angle APD$.

**Problem 3: 5** Find all prime numbers $p, q, r$, such that $\frac{p}{q} - \frac{4}{r+1} = 1$.

**Problem 4: 6** A $4 \times 4$ table is divided into 16 white unit square cells. Two cells are called neighbors if they share a common side. A move consists in choosing a cell and changing the colors of neighbors from white to black or from black to white. After exactly $n$ moves all the 16 cells were black. Find all possible values of $n$.

### Problem 1/4: 5 There are $a+b$ bowls arranged in a row, numbered 1 through $a+b$, where $a$ and $b$ are given positive integers. Initially, each of the first $a$ bowls contains an apple, and each of the last $b$ bowls contains a pear. A legal move consists of moving an apple from bowl $i$ to bowl $i+1$ and a pear from bowl $j$ to bowl $j-1$, provided that the difference $i-j$ is even. We permit multiple fruits in the same bowl at the same time. The goal is to end up with the first $b$ bowls each containing a pear and the last $a$ bowls each containing an apple. Show that this is possible if and only if the product $ab$ is even.

**Problem 2/5: 6-6.5** Let $a, b, c$ be positive real numbers such that $a + b + c = 4\sqrt[3]{abc}$. Prove that
$$2(ab + bc + ca) + 4\min(a^2, b^2, c^2) \geq a^2 + b^2 + c^2.$$

**Problem 3/6: 7** Two rational numbers $\frac{m}{n}$ and $\frac{n}{m}$ are written on a blackboard, where $m$ and $n$ are relatively prime positive integers. At any point, Evan may pick two of the numbers $x$ and $y$ written on the board and write either their arithmetic mean $\frac{x+y}{2}$ or their harmonic mean $\frac{2xy}{x+y}$ on the board as well. Find all pairs $(m, n)$ such that Evan can write 1 on the board in finitely many steps.

### HMMT (February) Individual Round, Problem 1-5: 5 Individual Round, Problem 6-10: 5.5-6 Team Round: 7.5 HMIC: 8

### APMO Problem 1: 6 Problem 2: 7 Problem 3: 7 Problem 4: 7.5 Problem 5: 8.5

### Balkan MO Problem 1: 5 Solve the equation $3^x - 5^y = z^2$ in positive integers. Problem 2: 6.5 Let $MN$ be a line parallel to the side $BC$ of a triangle $ABC$, with $M$ on the side $AB$ and $N$ on the side $AC$. The lines $BN$ and $CM$ meet at point $P$. The circumcircles of triangles $BMP$ and $CNP$ meet at two distinct points $P$ and $Q$. Prove that $\angle BAQ = \angle CAP$. Problem 3: 7.5 A $9 \times 12$ rectangle is partitioned into unit squares. The centers of all the unit squares, except for the four corner squares and eight squares sharing a common side with one of them, are coloured red. Is it possible to label these red centres $C_1, C_2..., C_{96}$ in such way that the following to conditions are both fulfilled (i) the distances $C_1C_2, ...C_{95}C_{96}, C_{96}C_1$ are all equal to $\sqrt{13}$ (ii) the closed broken line $C_1C_2...C_{96}C_1$ has a centre of symmetry? Problem 4: 8 Denote by $S$ the set of all positive integers. Find all functions $f : S \rightarrow S$ such that

$$f\left( f^2(m) + 2f^2(n) \right) = m^2 + 2n^2 \text{ for all } m, n \in S.$$

### USAMO Problem 1/4: 6-7 Problem 2/5: 7-8 Three nonnegative real numbers $r_1, r_2, r_3$ are written on a blackboard. These numbers have the property that there exist integers $a_1, a_2, a_3$, not all zero, satisfying $a_1r_1 + a_2r_2 + a_3r_3 = 0$. We are permitted to perform the following operation: find two numbers $x, y$ on the blackboard with $x \leq y$, then erase $y$ and write $y - x$ in its place. Prove that after a finite number of such operations, we can end up with at least one 0 on the blackboard. Problem 3/6: 8-9 Prove that any monic polynomial (a polynomial with leading coefficient 1) of degree $n$ with real coefficients is the average of two monic polynomials of degree $n$ with $n$ real roots.

### USA TST Problem 1/4/7: 6.5-7 Problem 2/5/8: 7.5-8 Problem 3/6/9: 8.5-9

### Putnam Problem A/B,1-2: 7 Find the least possible area of a concave set in the 7-D plane that intersects both branches of the hyperparabola $xyz = 1$ and both branches of the hyperbola $xwy = -1$. (A set $S$ in the plane is called convex if for any two points in $S$ the line segment connecting them is contained in $S$.) Problem A/B,3-4: 8 Let $H$ be an $n \times n$ matrix all of whose entries are $\pm 1$ and whose rows are mutually orthogonal. Suppose $H$ has an $a \times b$ submatrix whose entries are all 1. Show that $ab \leq n$. Problem A/B,5-6: 9 For any $a > 0$, define the set $S(a) = \{[an] | n = 1, 2, 3, ...\}$. Show that there are no three positive reals $a, b, c$ such that $S(a) \cap S(b) = S(b) \cap S(c) = S(c) \cap S(a) = \emptyset, S(a) \cup S(b) \cup S(c) = \{1, 2, 3, ...\}$.

### China TST (hardest problems) Problem 1/4: 8-8.5 Given an integer $m$, prove that there exist odd integers $a, b$ and a positive integer $k$ such that

$$2m = a^{19} + b^{99} + k * 2^{1000}.$$

Problem 2/5: 9 Given a positive integer $n = 1$ and real numbers $a_1 < a_2 < \ldots < a_n$, such that $\dfrac{1}{a_1} + \dfrac{1}{a_2} + \ldots + \dfrac{1}{a_n} \leq 1$, prove that for any positive real number $x$,

$$\left( \frac{1}{a_1^2 + x} + \frac{1}{a_2^2 + x} + \ldots + \frac{1}{a_n^2 + x} \right)^2 \geq \frac{1}{2a_1(a_1 - 1) + 2x}.$$

Problem 3/6: 9.5-10 Let $n > 1$ be an integer and let $a_0, a_1, \ldots, a_n$ be non-negative real numbers. Define $S_k = \sum_{i=0}^{k} \binom{k}{i} a_i$ for $k = 0, 1, \ldots, n$. Prove that

$$\frac{1}{n} \sum_{k=0}^{n-1} S_k^2 - \frac{1}{n^2} \left( \sum_{k=0}^{n} S_k \right)^2 \leq \frac{4}{45} (S_n - S_0)^2.$$

### IMO **Problem 1/4: 5.5-7** Let $\Gamma$ be the circumcircle of acute triangle $ABC$. Points $D$ and $E$ are on segments $AB$ and $AC$ respectively such that $AD = AE$. The perpendicular bisectors of $BD$ and $CE$ intersect minor arcs $AB$ and $AC$ of $\Gamma$ at points $F$ and $G$ respectively. Prove that lines $DE$ and $FG$ are either parallel or they are the same line.

**Problem 2/5: 7-8** Let $P(x)$ be a polynomial of degree $n > 1$ with integer coefficients, and let $k$ be a positive integer. Consider the polynomial $Q(x) = P(P(\ldots P(P(x))\ldots))$, where $P$ occurs $k$ times. Prove that there are at most $n$ integers $t$ such that $Q(t) = t$.

**Problem 3/6: 9-10** Let $ABC$ be an equilateral triangle. Let $A_1, B_1, C_1$ be interior points of $ABC$ such that $BA_1 = A_1C$, $CB_1 = B_1A$, $AC_1 = C_1B$, and

$$\angle BA_1C + \angle CB_1A + \angle AC_1B = 480°$$

Let $BC_1$ and $CB_1$ meet at $A_2$, let $CA_1$ and $AC_1$ meet at $B_2$, and let $AB_1$ and $BA_1$ meet at $C_2$. Prove that if triangle $A_1B_1C_1$ is scalene, then the three circumcircles of triangles $AA_1A_2$, $BB_1B_2$ and $CC_1C_2$ all pass through two common points.

### IMO Shortlist

Problem 1-2: 5.5-7

Problem 3-4: 7-8

Problem 5+: 9-10

</difficulty reference>

# OBJECTIVE #

1. Summarize the math problem in a brief sentence, describing the concepts involved in the math problem.

2. Based on the source of the given problem, as well as the difficulty of the problems referenced in these materials and the solution to the current problem, please provide an overall difficulty score for the current problem. The score should be a number between 1 and 10, with increments of 0.5, and should align perfectly with the materials.

# STYLE #

Data report.

# TONE #

Professional, scientific.

# AUDIENCE #

Students. Enable them to better understand the difficulty of the math problems.

# RESPONSE: MARKDOWN REPORT #

## Summarization

[Summarize the math problem in a brief paragraph.]

## Difficulty

[Rate the difficulty of the math problem and give the reason.]

# ATTENTION #

- Add "=== report over ===" at the end of the report.

<example math problem>

[Question]:

If $\frac{1}{9} + \frac{1}{18} = \frac{1}{x}$, what is the number that replaces the $x$ to make the equation true?

[Solution]:

We simplify the left side and express it as a fraction with numerator 1: $\frac{1}{9} + \frac{1}{18} = \frac{2}{18} + \frac{1}{18} = \frac{3}{18} = \frac{1}{6}$. Therefore, the number that replaces the $\square$ is 6.

[Source]: 2010_Pascal

</example math problem>

## Summarization

The problem requires finding a value that makes the equation $\frac{1}{9} + \frac{1}{18} = \frac{1}{\square}$. This involves adding two fractions and determining the equivalent fraction.

## Difficulty

Rating: 1

Reason: This problem is straightforward and primarily involves basic fraction addition, making it suitable for early middle school students.

=== report over ===

<example math problem>

[Question]:

Let $\mathcal{P}$ be a convex polygon with $n$ sides, $n \geq 3$. Any set of $n-3$ diagonals of $\mathcal{P}$ that do not intersect in the interior of the polygon determine a triangulation of $\mathcal{P}$ into $n-2$ triangles. If $\mathcal{P}$ is regular and there is a triangulation of $\mathcal{P}$ consisting of only isosceles triangles, find all the possible values of $n$.

[Solution]:

We label the vertices of $\mathcal{P}$ as $P_0, P_1, P_2, \ldots, P_n$. Consider a diagonal $d = \overline{P_a\, P_{a+k}}$, $k \leq n/2$ in the triangulation. We show that $k$ must have the form $2^m$ for some nonnegative integer $m$. This diagonal partitions $\mathcal{P}$ into two regions $\mathcal{Q}, \mathcal{R}$, and is the side of an isosceles triangle in both regions. Without loss of generality suppose the area of $Q$ is less than the area of $R$ (so the center of $P$ does not lie in the interior of $Q$); it follows that the lengths of the edges and diagonals in $Q$ are all smaller than $d$. Thus $d$ must the be the base of the isosceles triangle in $Q$, from which it follows that the isosceles triangle is $\triangle P_a P_{a+k/2} P_{a+k}$, and so $2|k$. Repeating this process on the legs of isosceles triangle $(\overline{P_a P_{a+k/2}}, \overline{P_{a+k} P_{a+k/2}})$, it follows that $k = 2^m$ for some positive integer $m$ (if we allow degeneracy, then we can also let $m = 0$). Now take the isosceles triangle $P_x P_y P_z$, $0 \leq x < y < z < n$ in the triangulation that contains the center of $\mathcal{P}$ in its interior; if a diagonal passes through the center, select either of the isosceles triangles with that diagonal as an edge. Without loss of generality, suppose $P_x P_y = P_y P_z$. From our previous result, it follows that there are $2^a$ edges of $P$ on the minor arcs of $P_x P_y$, $P_y P_z$ and $2^b$ edges of $P$ on the minor arc of $P_z P_x$, for positive integers $a$, $b$. Therefore, we can write

$$n = 2 \cdot 2^a + 2^b = 2^{a+1} + 2^b,$$

so $n$ must be the sum of two powers of 2. We now claim that this condition is sufficient. Suppose without loss of generality that $a + 1 \geq b$; then we rewrite this as

$$n = 2^b(2^{a-b+1} + 1).$$

Lemma 1: All regular polygons with $n = 2^k + 1$ or $n = 4$ have triangulations that meet the conditions. By induction, it follows that we can cover all the desired $n$. For $n = 3, 4$, this is trivial. For $k > 1$, we construct the diagonals of equal length $\overline{P_0 P_{2^{k-1}}}$ and $\overline{P_{2^{k-1}+1} P_0}$. This partitions $\mathcal{P}$ into 3 regions: an isosceles $\triangle P_0 P_{2^{k-1}} P_{2^{k-1}+1}$, and two other regions. For these two regions, we can recursively construct the isosceles triangles defined above in the second paragraph. It follows that we have constructed $2(2^{k-1} - 1) + (1) = 2^k - 1 = n - 2$ isosceles triangles with non-intersecting diagonals, as desired. Lemma 2: If a regular polygon with $n$ sides has a working triangulation, then the regular polygon with $2n$ sides also has a triangulation that meets the conditions. We construct the diagonals $\overline{P_0 P_2}$, $\overline{P_2 P_4}$, $\ldots \overline{P_{2n-2} P_0}$. This partitions $\mathcal{P}$ into $n$ isosceles triangles of the form $\triangle P_{2k} P_{2k+1} P_{2k+2}$, as well as a central regular polygon with $n$ sides. However, we know that there exists a triangulation for the $n$-sided polygon that yields $n-2$ isosceles triangles. Thus, we have created $(n) + (n-2) = 2n - 2$ isosceles triangles with non-intersecting diagonals, as desired. In summary, the answer is all $n$ that can be written in the form $2^{a+1} + 2^b$, $a, b \geq 0$. Alternatively, this condition can be expressed as either $n = 2^k$, $k \geq 2$ (this is the case when $a + 1 = b$) or $n$ is the sum of two distinct powers of 2, where $1 = 2^0$ is considered a power of 2.

[Source]:

USAMO 2008

</example math problem>

## Summarization

The problem asks for the possible values of $n$ for a regular n-sided polygon that can be completely triangulated into isosceles triangles using non-intersecting diagonals. The solution involves analyzing the properties of the diagonals forming isosceles triangles and deducing that $n$ can be expressed in terms of powers of 2.

## Difficulty

Rating: 7

Reason: The problem involves understanding properties of isosceles triangles in the context of polygon triangulation and requires critical reasoning to establish relationships between the number of sides and powers of 2, making it more complex than typical undergraduate-level problems.

=== report over ===

<math problem>
[QUESTION]:
***Question***
[SOLUTION]:
***Solution***
[SOURCE]:
***SOURCE***
</math problem>

Your answer should be in JSON format for example:
```json
{
"Rating": YOUR RATING,
"Reason": YOUR JUSTIFICATION,
}
```

## F.7 TOPIC DOMAIN

### Meta Prompt for Topic Domain Annotation

I am a teacher, and I have some high-level Olympiad math problems.
I want to categorize the domain of these math problems.

# OBJECTIVE #
1. Summarize the math problem in a brief sentence, describing the concepts involved in the math problem.
2. Categorize the math problem into specific mathematical domains. Please provide a classification chain, for example, Applied Mathematics -> Probability -> Combinations.
The following is a basic classification framework in the field of mathematics.
<math domains>
...
</math domains>
# STYLE #
Data report.
# TONE #
Professional, scientific.
# AUDIENCE #
Students. Enable them to better understand the domain of the problems.
# RESPONSE: MARKDOWN REPORT #
## Summarization
[Summarize the math problem in a brief paragraph.]
## Math domains
[Categorize the math problem into specific mathematical domains, including major domains and subdomains.]
# ATTENTION #
- The math problem can be categorized into multiple domains, but no more than three. Separate the classification chains with semicolons(;).
- Your classification MUST fall under one of the aforementioned subfields; if it really does not fit, please add "Other" to the corresponding branch. For example: Algebra -> Intermediate Algebra -> Other. Only the LAST NODE is allowed to be "Other"; the preceding nodes must strictly conform to the existing framework.
- The math domain must conform to a format of classification chain, like "Applied Mathematics -> Probability -> Combinations".

- Add "=== report over ===" at the end of the report.

\<example math problem\>.
[Question]
Determine the greatest real number $C$, such that for every positive integer $n \geq 2$, there exists $x_1, x_2, ..., x_n \in [-1, 1]$, so that

$$\prod_{1 \leq i < j \leq n} (x_i - x_j) \geq C^{\frac{n(n-1)}{2}}$$

\</example math problem\>
## Summarization
The problem seeks to find the greatest real number $C$ such that, for every integer $n \geq 2$, there exist real numbers $x_1, x_2, \ldots, x_n \in [-1, 1]$ satisfying the inequality $\prod_{1 \leq i < j \leq n} (x_i - x_j) \geq C^{\frac{n(n-1)}{2}}$. This involves maximizing $C$ to ensure the product of all pairwise differences among $n$ points in the interval $[-1, 1]$ is at least $C$ raised to the power of the number of such pairs, $\frac{n(n-1)}{2}$.
## Math domains
Algebra -> Intermediate Algebra -> Inequalities; Discrete Mathematics -> Combinatorics
=== report over ===

\<example math problem\>
[Question]
Given integer $n \geq 2$. Find the minimum value of $\lambda$, satisfy that for any real numbers $a_1, a_2, \cdots, a_n$ and $b$,

$$\lambda \sum_{i=1}^{n} \sqrt{|a_i - b|} + \sqrt{n \left| \sum_{i=1}^{n} a_i \right|} \geqslant \sum_{i=1}^{n} \sqrt{|a_i|}.$$

\</example math problem\>
## Summarization
Let $n \geq 2$ be an integer. The problem seeks the minimum value of $\lambda$ such that for any real numbers $a_1, a_2, \ldots, a_n$ and $b$, the inequality $\lambda \sum_{i=1}^{n} \sqrt{|a_i - b|} + \sqrt{n |\sum_{i=1}^{n} a_i|} \geqslant \sum_{i=1}^{n} \sqrt{|a_i|}$ holds. The goal is to find the smallest $\lambda$ that ensures this inequality is satisfied for all possible choices of $a_i$ and $b$.
## Math domains
Algebra -> Intermediate Algebra -> Inequalities;
Calculus -> Differential Calculus -> Applications of Derivatives.
=== report over ===

\<math problem\>
[Question]
***Question***
[Solution]
***Solution***
[Source]
***Source***
\</math problem\>

Your answer should be in JSON format for example:
""" json
{
"Summary": "YOUR_SUMMARY",
"Domains": [domain1, ...]
}
"""

## F.8 THEOREM VARIANT CONSTRUCTION

---

**Meta Prompt for Constructing Theorem Variants**

I'm going to give you a math proof question and its solution. Your task is to follow the steps below to write a new question based on the given one. Here is the original question and solution:

```question
<question>
```,
```solution
<solution>
```

Please follow these steps:
1. The original question asks to "prove or disprove" a statement, where the statement can be "proved". Please write a new question by negating the original statement, so that it can now be "disproved". For example, if the original statement is $x = y$, you may change it to $x < y$ or $x \neq y$; if the statement is "there exists xxx", you may change it to "there does not exist xxx". When negating the original question, you should make minimal changes, i.e. leave as much background information unchanged as possible.

2. After changing the question, the solution should be changed accordingly. You do not have to write a new solution, and the original solution can probably be reused. For example, if the original question asks to prove $x = y$ and the new question asks to prove $x < y$, you may simply add a step to the original proof like "since we proved $x = y$, the statement $x < y$ is disproved". However, check the wording of the solution so that it tries to "prove" the statement at first, and then naturally transit to finding that it cannot be proved, but can be disproved instead.

3. The original solution ends with "\boxed{proved}". Your new solution should end with "\boxed{disproved}".
Output the new question and solution in two blocks:
```question
new question
```
```solution
corresponding solution
```

---

## F.9 SYSTEM PROMPT FOR RL TRAINING

---

**System Prompt for RL Training**

A conversation between User and Assistant. The User gives a statement, and the Assistant either proves or disproves it. The Assistant first thinks about the reasoning process in the mind and then provides the User with the answer. The reasoning process is enclosed within <think> </think> and the answer is enclosed within <answer> </answer> tags, respectively, i.e., <think> reasoning process here </think> <answer> answer here </answer>. If you prove the statement, answer with "proved". If you disprove the statement, answer with "disproved". You must put your answer inside <answer> </answer> tags, i.e., <answer> \boxed{proved} or \boxed{disproved} </answer>. And your final answer will be extracted automatically by the \boxed{} tag.

---

## F.10 PROCESS EVALUATION FRAMEWORK

---

**Meta Prompt for Process Evaluation**

You are an expert in scoring solutions for mathematical proof questions. The following question asks to prove or disprove a statement, where the statement may be either true or false. The test subject is asked to end their proof with \boxed{proved} if they prove the statement to be true, and \boxed{disproved} if they prove the statement to be false.

The question:
"'<question>"'
The ground truth of the statement:
"'<answer>"'
The test subject's solution:
"' <solution>"'

Your task is to evaluate the proof's quality and assign a score from 0 to 1 based on four criteria: logical validity (40%), completeness (30%), correctness (20%), and clarity (10%).
Instructions:
1. Analyze the proof step by step.
2. For each criterion:
- Logical Validity: Check if each step follows logically from the previous one. Flag any logical errors.
- Completeness: Verify if all necessary cases and steps are included to prove the theorem.
- Correctness: Confirm if the final conclusion is correct.
- Clarity: Assess if the proof is clear, unambiguous, and well-explained.
3. Assign a sub-score (0 to 1) for each criterion and compute the total score using the weights: $(0.4 \times$ validity$) + (0.3 \times$ completeness$) + (0.2 \times$ correctness$) + (0.1 \times$ clarity$)$.
4. Provide a brief explanation (2-3 sentences) summarizing any errors or issues and justifying the score.
Final output format:
"'

{
"score": float,
"validity": float,
"completeness": float,
"correctness": float,
"clarity": float,
"explanation": str
}
"'

where "score" is the total score, and "validity", "completeness", "correctness", "clarity" are the subscores.

---

