# OpenReview forum: "DeepTheorem: Advancing LLM Reasoning for Theorem Proving Through Natural Language and Reinforcement Learning"
_ICLR.cc/2026/Conference — Submitted to ICLR 2026_

### Official Review · Reviewer_Ua9x · 2025-10-25

**Soundness:** 2
**Presentation:** 3
**Contribution:** 2
**Rating:** 2
**Confidence:** 4

**Summary:**

The paper introduces DeepTheorem, a suite for informal (natural‑language/LaTeX) theorem proving consisting of a 121K-sample dataset of IMO‑level natural‑language theorems with concise proof sketches, topic labels, and a family of truth‑preserving or contradictory variants per theorem to enable automatic checking; 2) an adaptation of RL‑Zero/GRPO to informal theorem proving that uses pass/fail signals derived from those variants as rewards; and 3) an evaluation framework with both outcome accuracy (truth classification across variants) and process scoring of proofs by LLM judges, with a small human study for calibration

**Strengths:**

1. Data Scale:  The 121K IMO‑level, topic‑labeled problems with curated entailed/contradictory variants are materially larger than prior NL theorem/proof corpora (e.g., NaturalProofs/NaturalProver focused on smaller‑scale NL proofs). The variant family provides a neat mechanism for binary checking without a formal verifier, and Figure 4 (p.4) shows difficulty calibration that overlaps harder benchmarks (miniF2F, FIMO, Putnam, HMMT).

2. Empirical gains: Table 3 (p.7) and Figure 6 (p.8) indicate consistent improvements of RL‑Zero over SFT across Qwen2.5 7B/72B models on the proposed outcome‑metric and process‑scores.

**Weaknesses:**

1. The dataset is largely curated via web scraping existing problem sets and multiple LLM passes, with the final “step-by-step proofs” produced by black-box commercial models o3-mini. The RL part applies the naive 1/0 correctness reward extracted; and the process evaluation itself is performed by a black-box LLM judge (GPT-4o, with o3-mini as an alternative) rather than formal proof assistants such as lean. This pipeline raises concerns about originality (vs. prior RL-for-reasoning recipes), circularity, and reproducibility.

2. Out-of-date baselines. The core experiments fine-tune Qwen2.5 model families and compare against a mix of 2024 models; however, several contemporaneous stronger families (e.g., Qwen3 dense / MoE series, GPT-5 ) and post-2024 formal provers (e.g., Seed-Prover variants) have significantly improved the frontier. Using pre-Qwen3 models limits the timeliness of the claims and their competitiveness at submission time.

3. Unfair comparisons with formal theorem proving baselines. The test sets on table 4 from FIMO and PutnamBench are formal theorem-proving benchmarks but are manually converted to informal, variant-based NL evaluation; results are then compared against models for formal-theorem proving (such as DeepSeek-Prover) with formal accuracy (much more difficult than informal accuracies)Comparing an informal truth-assignment outcome to formal proof success under a verifier is not methodologically fair even if both are reported on the same underlying source problems.

4. Decontamination is not documented w.r.t. time via temporal split. The paper details an embedding-based recall-and-justify decontamination and shows removed examples, but it does not report any timestamp/knowledge-cutoff filtering (e.g., train < test release date), nor quantitative false-positive/false-negative audits. Without a temporal split, leakage from recent competitions and public solutions remains a risk.

5. Tiny human eval: the process scores depend on black-box GPT-4o/o3-mini judgments. While the paper reports high correlation between judges, LLM-based grading cannot ensure semantic correctness the way a formal checker can, and the human study covers only 10 items with two raters, which is far too small to validate reliability.

**Questions:**

See weakness

---

> ### Author Response · Authors · 2025-11-23
> **Response to Reviewer Ua9x, part 1**
>
> We sincerely thank the reviewer for raising these insightful questions, which have helped us substantially refine both the clarity and rigorousness of our work. We provide the following responses to summarize our revisions and clarify any potential misunderstanding about our work. Unless otherwise specified, all section numbers, line numbers, tables, and figures below refer to the revised paper.
>
> ### 1. Relation to Formal Theorem Proving
>
> We would like to clarify a key point: **our paper focuses exclusively on informal theorem proving, and does not involve any experimental comparison with Lean4 or other formal engines**. In the initial submission, experiments with DS-Prover are conducted in the same setting as other LLMs (i.e. generating informal proofs).
>
> Recognizing the potential confusion, we have made the following revisions to align the narrative more closely with our contributions and eliminate any potential ambiguity:
>
> - We replaced Lean-WorkBook and DS-Prover datasets in Figure 1 and 4 with recently released informal theorem datasets, including OPC [1], IneqMath [2], and RFMDataset [3], all of which are designed for analytical purposes rather than model training and are thus much smaller than DeepTheorem.
>
> - We removed the two DS-Prover models from the main experiments in Table 4, and replaced them with three of the most recent and capable open-source models: Qwen3-32B [4], Kimi-K2-Thinking-1T [5], and Ling-1T [6].
>
> - We revised the first half of the related work section to minimize discussion of works on formal theorem proving and emphasize advances in informal theorem proving.
>
> [1] The Open Proof Corpus: A Large-Scale Study of LLM-Generated Mathematical Proofs. https://arxiv.org/abs/2506.21621
>
> [2] Solving Inequality Proofs with Large Language Models. https://arxiv.org/abs/2506.07927
>
> [3] Mathematical Proof as a Litmus Test: Revealing Failure Modes of Advanced Large Reasoning Models. https://arxiv.org/abs/2506.17114.
>
> [4] Qwen3 Technical Report. https://arxiv.org/abs/2505.09388
>
> [5] Kimi K2: Open Agentic Intelligence. https://arxiv.org/abs/2507.20534
>
> [6] Every Activation Boosted: Scaling General Reasoner to 1 Trillion Open Language Foundation. https://arxiv.org/abs/2510.22115
>
> ### 2. Reproducibility
>
> We fully agree that reproducibility is crucial to the development of the AI community. To this end, **we will publicly release all artifacts generated in this work**, including:
>
> - The full DeepTheorem training set (121K theorems, 242K variants);
> - Intermediate LLM annotations;
> - The evaluation benchmarks;
> - Training code;
> - Trained model checkpoints;
> - The newly collected human annotations (detailed below).
>
> Furthermore, **to ensure full reproducibility of our evaluation pipeline, we have additionally included an open-source LLM judge and an open process reward model in the revision** (detailed below).

---

> > ### Author Response · Authors · 2025-11-23
> > **Response to Reviewer Ua9x, part 2**
> >
> > ### 3. Qwen3 Baselines
> >
> > **We have included new training experiments using Qwen3-4B as the backbone in Appendix C, training four sets of models on 100K-sample subsets to ensure fair comparison**: (1) the baseline OpenR1-Proof dataset, (2) the DeepTheorem dataset, (3) easy samples (level 5-7) in DeepTheorem, (4) hard samples (level 7-9) in DeepTheorem. The results are presented in Table 8. We find that
> >
> > 1. **SFT is more effective for the (relatively) easy data, while RL is more effective for the hard data**;
> > 2. **simply combining the easy and hard data does not lead to improved performance**;
> > 3. We also provide the results of training Qwen3-4B on the full DeepTheorem dataset in Table 9, showing that **scaling to the full dataset mitigates mixture drawbacks and consistently boosts results, highlighting the practical value of DeepTheorem as a large-scale training set**;
> > 4. **There is no significant correlation between performance and generation lengths**. On the sampled data (Table 8), SFT models consistently exhibit longer generation lengths, while on the full data similar generation lengths are observed.
> >
> > Besides the training experiments, **we also added three most advanced open-source LLMs to the LLM evaluation in Table 4** - Qwen3-32B, Kimi-K2-Thinkg-1T, and Ling-1T - to ensure that our benchmarks reflect the current state of open-source models.
> >
> > ### 4. LLM Judge Reliability Analysis
> >
> > We acknowledge the inherent limitations of LLM-as-a-judge evaluation. To address this, we have substantially expanded our validation of the process evaluation methodology along three complementary aspects.
> >
> > First, instead of relying solely on API-based models, **we evaluated a third, open-source LLM - Ling-1T**, one of the largest and most capable open-source models. The results are added to Table 10 in the appendix, which **exhibit a high consistency with the closed-source models**, with a correlation coefficient of more than 0.98 with both GPT-4o and o3-mini.
> >
> > Second, **we also incorporated a comparison with existing process evaluation models (Table 10)**, specifically Qwen2.5-Math-PRM [7], which scores every reasoning step in the model solutions (delimited by double line breaks) without requiring a dedicated prompt. **The results also align well with all three LLM judges, further verifying the robustness of our evaluation protocol**. The full correlation matrix is given below:
> >
> > |         | GPT-4o | o3-mini | Ling-1T | Q2.5-PRM-72B |
> > | ------- | ------ | ------- | ------- | ------------ |
> > | GPT-4o  | -      | 0.9545  | 0.9872  | 0.9964       |
> > | o3-mini | -      | -       | 0.9866  | 0.9477       |
> > | Ling-1T | -      | -       | -       | 0.9824       |
> >
> > Third, **we expanded the human evaluation** by recruiting 12 third-party expert annotators (all graduate students majoring in mathematics) to perform pairwise comparisons of model responses across 100 questions and 10 models. **The results (Table 11) are highly consistent with preferences computed from the scores of LLM judges, demonstrating strong external validity**. We will release these human annotations along with other artifacts created in this research.
> >
> > [7] The Lessons of Developing Process Reward Models in Mathematical Reasoning. https://aclanthology.org/2025.findings-acl.547/
> >
> > ### 5. Decontamination
> >
> > As detailed in Line 160-182, **our training data has undergone rigorous decontamination against a comprehensive set of mathematical reasoning and theorem-proving benchmarks**. Our decontamination pipeline follows the recall-and-justify methodology introduced in the seminal work OpenMathInstruct-2 [8], which combines embedding-based similarity search with LLM-based justification to remove any lexical or semantical overlap with 20+ established benchmarks (including those used for evaluation in our work). **This process removed almost 200K samples (Figure 5), including identical, generalized, and logically converse cases (examples in Table 7)**. Moreover, **our evaluation benchmarks consist of manually expanded theorem variants that are not present in any existing dataset, ensuring the integrity of our evaluation pipeline**.
> >
> > [8] OpenMathInstruct-2: Accelerating AI for Math with Massive Open-Source Instruction Data. https://arxiv.org/abs/2410.01560
> >
> > Once again, we are deeply grateful for the reviewer’s rigorous feedback. We believe the revisions - especially the expanded training experiments, process evaluation validation, and emphasis on the positioning of our paper - have substantially strengthened the paper's technical soundness, reproducibility, and alignment with the informal theorem proving literature.

---

### Official Review · Reviewer_zq5f · 2025-10-25

**Soundness:** 1
**Presentation:** 3
**Contribution:** 2
**Rating:** 2
**Confidence:** 4

**Summary:**

This paper introduces DeepTheorem, a large-scale dataset designed for informal theorem proving. The dataset is richly annotated with information such as difficulty levels, proof labels, topic categories, problem variants, and exampled proofs. Building on this resource, the authors adapt RL-Zero to train large language models on DeepTheorem and propose both outcome-based and process-based evaluation metrics, using LLMs as judges. Experimental results demonstrate that models trained on DeepTheorem achieve stronger performance on external theorem-proving benchmarks compared to baselines of the same model scale.

**Strengths:**

* The paper is overall well-written and easy to follow, with an interesting focus on informal theorem proving.
* The proposed dataset is large-scale and carefully annotated with detailed information such as problem variants and difficulty levels.
* The LLM trained on this dataset achieves better performance than other baselines of the same model scale.

**Weaknesses:**

I think the major weakness of this paper lies in the fact that the entire dataset construction and evaluation pipeline is based solely on LLMs.

* First, the original problem statements are drawn from existing datasets, which may have been included in the pretraining data of some LLMs. This raises concerns about potential contamination and data leakage.

* Second, the generation of variants, the labeling of True or False, the construction of proofs, and even the assignment of difficulty levels and topics all depend heavily on existing LLMs—particularly o3-mini. In this sense, the dataset can be seen as a form of distillation of o3-mini onto a new dataset, rather than a fully independent benchmark.

* Third, there is no human evaluation of the LLM annotations—not even on a sampled subset—to assess their quality. This absence raises serious concerns about the correctness of the proofs, labels, and overall dataset integrity.

* Moreover, the evaluation relies on GPT-4o and o3-mini as LLM-as-a-judge models, with no manual inspection, even for a small portion of the data. Prior work has shown that such models may produce biased or inflated scores toward certain stylistic outputs (e.g., their own) [1]. Therefore, using a single LLM for evaluation, without any human verification, makes the results less reliable and less sound.

* Finally, the baseline dataset appears to contain much less data, making the comparison seem somewhat unfair and potentially misleading.

Minor: The question shown in Figure 2 appears incomplete, which further raises concerns about the dataset’s quality and attention to detail.

[1] Gu, Jiawei, et al. “A Survey on LLM-as-a-Judge.” arXiv preprint arXiv:2411.15594 (2024).

**Questions:**

Could you manually evaluate a subset of the dataset to assess the accuracy and reliability of both the LLM-generated annotations and the LLM-as-a-judge evaluations?

---

> ### Author Response · Authors · 2025-11-23
> **Response to Reviewer zq5f, part 1**
>
> We sincerely thank the reviewer for the thoughtful and incisive feedback, which has helped us significantly strengthen the rigor and soundness of our work. Unless otherwise specified, all section numbers, line numbers, tables, and figures below refer to the revised paper.
>
> ### 1. Data Contamination
>
> As detailed in Line 160-182, **our training data has undergone rigorous decontamination against a comprehensive set of mathematical reasoning and theorem-proving benchmarks**. While it is true that the pretraining corpora of most advanced LLMs are not publicly disclosed - making complete decontamination impractical - we emphasize that **the DeepTheorem dataset is designed as a training resource, not an evaluation benchmark**. Therefore, potential overlap with pretraining data does not compromise the validity of our downstream experiments or conclusions, as all evaluations are conducted on held-out, manually verified test sets (FIMO, HMMT, Putnam) that are explicitly excluded from training.
>
> ### 2. Proof Generation and Variant Construction
>
> We acknowledge the reviewer’s observation that proof generation via o3-mini constitutes a form of knowledge distillation. This approach is **standard practice in current large-scale mathematical dataset construction**, as exemplified by OpenR1-Math (https://huggingface.co/open-r1). Besides, given the high reasoning cost of our teacher model (o3-mini) and the large scale of our dataset (121K), we expect the distillation itself to also represent a valuable contribution to the community.
>
> Importantly, **our variant generation procedure (Section 3.1) is deterministically grounded in mathematical logic**: each variant is constructed via minimal, verifiable transformations (e.g., sign flips, quantifier swaps) that preserve or invert logical entailment. This ensures that the reward signal used in RL is **objectively defined and mathematically unambiguous**, independent of any model’s internal reasoning.
>
> ### 3. Quality Control
>
> **The DeepTheorem dataset was constructed through a strict quality control pipeline that combines automated filtering with human-in-the-loop validation**. After each major processing step, we conducted manual inspections on random subsets to ensure fidelity and correctness.
>
> Moreover, the employment of LLMs for certain annotation steps is both well-documented and grounded in precedent. **For difficulty annotation, we adopt the same framework from the seminal work Omni-Math [1]. For decontamination, we took inspiration from OpenMathInstruct-2 [2].** To increase transparency, we include examples of the three representative contamination types - identical generalizing, and logically converse - in Table 7 (formerly Table 5). Additionally, we have now also included examples from different difficulty levels in Table 6 to offer direct insight into the dataset's rigor and diversity.
>
> [1] Omni-MATH: A Universal Olympiad Level Mathematic Benchmark For Large Language Models. https://arxiv.org/abs/2410.07985
>
> [2] OpenMathInstruct-2: Accelerating AI for Math with Massive Open-Source Instruction Data. https://arxiv.org/abs/2410.01560

---

> ### Author Response · Authors · 2025-11-23
> **Response to Reviewer zq5f, part 2**
>
> ### 4. Process Evaluation Methodology
>
> We fully recognize the inherent limitations of LLM-as-a-judge evaluation. To address this, we have rigorously expanded our validation of the process evaluation methodology along three complementary.
>
> First, instead of relying solely on API-based models, **we evaluated a third, open-source LLM - Ling-1T [3]**, one of the largest and most capable open-source models. The results are added to Table 10 in the appendix, which **exhibit a high consistency with the closed-source models**, with a correlation coefficient of more than 0.98 with both GPT-4o and o3-mini.
>
> Second, following the valuable suggestion from Reviewer EGQ7 **we also incorporated a comparison with existing process evaluation models**, specifically Qwen2.5-Math-PRM [3], which scores every reasoning step in the model solutions (delimited by double line breaks) without requiring a dedicated prompt. **The results (Table 10) also align well with all three LLM judges, further verifying the robustness of our evaluation protocol**. The full correlation matrix is given below:
>
> |         | GPT-4o | o3-mini | Ling-1T | Q2.5-PRM-72B |
> | ------- | ------ | ------- | ------- | ------------ |
> | GPT-4o  | -      | 0.9545  | 0.9872  | 0.9964       |
> | o3-mini | -      | -       | 0.9866  | 0.9477       |
> | Ling-1T | -      | -       | -       | 0.9824       |
>
> Third, **we expanded the human evaluation** by recruiting 12 third-party expert annotators (all graduate students majoring in mathematics) to perform pairwise comparisons of model responses across 100 questions and 10 models. **The results (Table 11) are highly consistent with preferences computed from the scores of LLM judges, demonstrating strong external validity**. We will release these human annotations along with other artifacts created in this research.
>
> [3] Every Activation Boosted: Scaling General Reasoner to 1 Trillion Open Language Foundation. https://arxiv.org/abs/2510.22115
>
> [4] The Lessons of Developing Process Reward Models in Mathematical Reasoning. https://aclanthology.org/2025.findings-acl.547/
>
> ### 5. Comparison with Baseline
>
> We sincerely thank the reviewer for this incisive comment. Combining this suggestion with the valuable feedback from Reviewer xMBg, **we supplemented a series of experiments in Appendix C, studying the impact of dataset size and sample difficulty**.
>
> Specifically, we trained four sets of models using Qwen3-4B backbone on four 100K-sample subsets: (1) the baseline OpenR1-Proof dataset, (2) the DeepTheorem dataset, (3) easy samples (level 5-7) in DeepTheorem, (4) hard samples (level 7-9) in DeepTheorem. The results are presented in Table 8. We find that
>
> 1. **SFT is more effective for the (relatively) easy data, while RL is more effective for the hard data**;
> 2. **simply combining the easy and hard data does not lead to improved performance**;
> 3. We also provide the results of training Qwen3-4B on the full DeepTheorem dataset in Table 9, showing that **scaling to the full dataset mitigates mixture drawbacks and consistently boosts results, highlighting the practical value of DeepTheorem as a large-scale training set**;
> 4. **There is no significant correlation between performance and generation lengths**. On the sampled data (Table 8), SFT models consistently exhibit longer generation lengths, while on the full data similar generation lengths are observed.
>
> ### 6. Incomplete Figure 2
>
> We sincerely appreciate the reviewer's attention to detail. The incompleteness in Figure 2 is not a defect in the dataset, but rather a paper typesetting oversight. This has been corrected in the revision.
>
> Once again, we are deeply grateful for the reviewer’s rigorous and constructive critique. We believe the revisions - particularly the expanded process evaluation validation and ablation studies - have substantially strengthened the paper's soundness.

---

### Official Review · Reviewer_xMBg · 2025-10-29

**Soundness:** 2
**Presentation:** 2
**Contribution:** 3
**Rating:** 4
**Confidence:** 3

**Summary:**

This paper presents DeepTheorem, a comprehensive framework for natural language theorem proving comprising four key components: 1) a large-scale dataset of 121K IMO-level informal theorems with high-quality proofs; 2) a novel adaptation of RL-Zero training methodology designed explicitly for informal theorem proving; 3) comprehensive outcome and process evaluation metrics for assessing proof quality; and 4) a newly constructed informal theorem proving benchmark with manually annotated theorem variants. This paper conducts experiments using both SFT and RL-Zero training methods on Qwen-2.5-Base across three sizes. Experiment results indicate that the model trained on DeepTheorem achieves leading performance compared to the baseline dataset. It also shows that the trained model obtains better results than other open-source models of similar size.

**Strengths:**

1. **Novel idea of verifying NL proofs using LLM:** The paper proposes a logically sound verification method to evaluate natural language theorem proving by following its 4-step criteria. The idea to prove/disprove the variants of the problem and combine all the results together to justify the correctness of the theorem is novel to the field.
2. **Leading benchmark results on open-source models:** The DeepTheorem dataset trained model achieves leading performance in open-source models and can even compete with large closed-source models. The experiment also thoroughly evaluates the difference between SFT and RL performance on different model sizes across different datasets to prove the effectiveness of DeepTheorem.
3. **Useful benchmark and dataset proposed:** The benchmark and dataset proposed in this paper are valuable for training more effective models. The scale of the dataset and soundness of the benchmark are all valuable contributions to the field.

**Weaknesses:**

Despite the above strengths, I have a few concerns regarding this paper:

1. **Unclear motivation and positioning:** This paper’s narrative is confusing regarding its claim of formal theorem proving. While this work focuses on informal theorem proving, the major comparison datasets shown in Figure 1 are DS-Prover-V1 and Lean-WB, both of which are based on Lean4 formal proving rather than informal reasoning. Similarly, the experiment section presents results from DS-Prover, which is not designed for informal reasoning. Besides, this paper also compares with Lean4 reasoning in many parts of the paper; however, the scope of the paper still compares the final outcome results, which differ from real formal theorem proving, where every step is verifiable. This creates confusion about the paper's positioning and contributions in relation to existing work on formal versus informal theorem proving.
2. **Insufficient experimental analysis:** The experimental evaluation lacks width and depth in the following ways:
    1. No ablation studies are provided to validate the effectiveness of different components of the proposed dataset (e.g., how training data at different difficulty levels affects model performance, what is the scaling-law effect for the dataset).
    2. The case study presents only examples without systematic analysis or insights.
    3. Critical training details are missing, including the exact number of GPU hours and computational costs.
    4. Given that RL-Zero is known for generating significantly longer responses, the paper should report generation length statistics, but this information is absent.
3. **Limited evaluation of generalizability:** The paper only evaluates DeepTheorem’s result on Qwen-2.5-Base across different sizes. It does not evaluate on additional model families such as Qwen-2.5-Math, Llama-4, and Qwen-3. It would be necessary to demonstrate the generalizability and robustness of the proposed dataset.
4. **Presentation quality issues:** Several figures suffer from poor quality or clarity problems. Specifically, Figures 3 and 7 contain blurred text and incomplete labels in the pie charts, making them difficult to interpret.

With the above weakness, I can only provide a relatively negative overall opinion of the paper. However, I am willing to increase the score if the raised concerns are properly settled.

**Questions:**

Here are the questions for the authors:

1. What is the connection between the proposed dataset and Lean-WB and DS-Prover-V1’s datasets? Is it necessary to compare them?
2. What is the ablation study of each component of the proposed method? Are there more case studies and analyses? What is the GPU cost for Zero-RL and SFT training for the models? And what is the generated content length as the training goes on?
3. How is the effect of DeepTheorem’s performance on SFT/RL on other leading open-source models?

---

> ### Author Response · Authors · 2025-11-23
> **Response to Reviewer xMBg, part 1**
>
> We sincerely thank the reviewer for recognizing the novelty, soundness, and valuable contribution of our paper. We are also deeply grateful for the thoughtful and constructive feedback, which has significantly helped us improve the paper. Unless otherwise specified, all section numbers, line numbers, tables, and figures below refer to the revised paper.
>
> ### 1. Motivation and Positioning
>
> We sincerely thank the reviewer for pointing out the potential confusion. We would like to emphasize that **our paper focuses exclusively on informal theorem proving and does not involve any experimental comparison with Lean4 or other formal engines**. The mentioning of Lean-Workbook and DS-Prover-V1 data is only meant as a background introduction to highlight the scarcity of informal theorem datasets, while experiments with DS-Prover in the initial submission are conducted in the same setting as other LLMs (i.e. generating informal proofs).
>
> To eliminate the confusion and ambiguity, we have made the following revisions to align the narrative more closely with our contributions:
>
> - We replaced Lean-WorkBook and DS-Prover datasets in Figure 1 and 4 with recently released informal theorem datasets, including OPC [1], IneqMath [2], and RFMDataset [3], all of which are designed for analytical purposes rather than model training and are thus much smaller than DeepTheorem.
>
> - We removed the two DS-Prover models from the main experiments in Table 4, and replaced them with three of the most recent and capable open-source models: Qwen3-32B [4], Kimi-K2-Thinking-1T [5], and Ling-1T [6].
>
> - We revised the first half of the related work section to minimize discussion of works on formal theorem proving and emphasize advances in informal theorem proving.
>
> [1] The Open Proof Corpus: A Large-Scale Study of LLM-Generated Mathematical Proofs. https://arxiv.org/abs/2506.21621
>
> [2] Solving Inequality Proofs with Large Language Models. https://arxiv.org/abs/2506.07927
>
> [3] Mathematical Proof as a Litmus Test: Revealing Failure Modes of Advanced Large Reasoning Models. https://arxiv.org/abs/2506.17114.
>
> [4] Qwen3 Technical Report. https://arxiv.org/abs/2505.09388
>
> [5] Kimi K2: Open Agentic Intelligence. https://arxiv.org/abs/2507.20534
>
> [6] Every Activation Boosted: Scaling General Reasoner to 1 Trillion Open Language Foundation. https://arxiv.org/abs/2510.22115

---

> ### Author Response · Authors · 2025-11-23
> **Response to Reviewer xMBg, part 2**
>
> ### 2. Experimental Analysis and Generalization
>
> We appreciate these constructive suggestions to strengthen experimental analysis, and have supplemented the following experiments and discussions.
>
> - For the **LLM evaluation in Table 4**, we have included three most recent and capable models as mentioned above (Qwen3-32B, Kimi-K2-Thinkg-1T, Ling-1T). Notably, we find Qwen3-32B and Ling-1T to perform exceptionally well on our theorem proving benchmarks.
>
> - For the **LLM judge evaluation in Appendix D**, we incorporated a third judge - the open-source Ling-1T model - and Qwen2.5-Math-PRM-72B, a dedicated process reward model following the valuable suggestion from Reviewer EGQ7. All four models exhibit high consistency in their scores, with pairwise correlation coefficients given below:
>   | | GPT-4o | o3-mini | Ling-1T | Q2.5-PRM-72B |
>   | ------- | ------ | ------- | ------- | ------------ |
>   | GPT-4o | - | 0.9545 | 0.9872 | 0.9964 |
>   | o3-mini | - | - | 0.9866 | 0.9477 |
>   | Ling-1T | - | - | - | 0.9824 |
>
> - We have also **expanded the human evaluation in Appendix D**, recruiting 12 third-party expert annotators (all graduate students majoring in mathematics) to conduct pairwise evaluations on 100 questions across 10 models. The results (Table 11) also show strong alignment with LLM judge rankings. We will release these human annotations along with other artifacts collected in this research.
>
> - For the **training experiments**, we supplemented results using Qwen3-4B as the backbone in Appendix C. On the full-scale Deeptheorem dataset (i.e. 242K variants from 121K theorems), training this model with SFT for 3 epochs takes 11 hours on 8 A100-80G GPUs, while training it with RL for 1 epoch takes 80 hours on 32 A100-80G GPUs. The results are presented in Table 9, demonstrating the effectiveness of RL over SFT.
>
> - Also using Qwen3-4B, we conducted **a series of ablation studies to explore the impact of data size and difficulty**. Specifically, we trained four sets of models on 100K theorem variants sampled from different data mixtures: (1) OpenR1-Proof, (2) DeepTheorem , (3) easy samples (level 5-7) in DeepTheorem, (4) hard samples (level 7-9) in DeepTheorem. The results are presented in Table 8. We find that
>
>   1. **SFT is more effective for the (relatively) easy data, while RL is more effective for the hard data**;
>   2. **simply combining the easy and hard data does not lead to improved performance**;
>   3. comparing Table 8 and Table 9, **scaling to the full dataset mitigates mixture drawbacks and consistently boosts results**;
>   4. **There is no significant correlation between performance and generation lengths**. On the sampled data (Table 8), SFT models consistently exhibit longer generation lengths, while on the full data similar generation lengths are observed.
>
> ### 3. Presentation Quality
>
> We sincerely thank the reviewer for the attention to detail. We have:
>
> - replaced both Figure 3 and 7 with vector graphs;
> - improved the layout in Figure 7 to incorporate all labels;
> - in the case of Figure 3, it is not possible to include all labels while maintaining readability due to the constrained space, so we have presented the complete topic breakdown as a separate table in Appendix A.
>
> Again, we sincerely thank the reviewer's insightful and constructive feedback. We believe the revisions have substantially strengthened the paper, and we hope our responses adequately address your concerns.

---

> > ### Comment · Reviewer_xMBg · 2025-11-26
> > **Further questions about your**
> >
> > Thank you so much for the rebuttal. The points I raised in Weakness 1 and 4, as well as Question 1, have been settled. However, it seems the author does respond to my other questions and concerns in the weaknesses section. My standing questions include:
> > 1. The case study with detailed analysis, analysis on why, under DeepTheorem training.
> > 2. The Zero-RL does not increase the generation length as other works [1] indicated.
> > 3. What is the performance of the dataset on other families of models?
> >
> > Including these experiments is vital for the paper's completeness and for evaluating the paper's contribution.
> >
> > **References:**
> >
> > [1] Guo, D., Yang, D., Zhang, H., Song, J., Zhang, R., Xu, R., ... & He, Y. (2025). Deepseek-r1: Incentivizing reasoning capability in llms via reinforcement learning. arXiv preprint arXiv:2501.12948.

---

> > > ### Author Response · Authors · 2025-12-02
> > >
> > > We are sincerely grateful for the reviewer's further feedback, and for emphasizing the importance of deeper analyses with case studies.
> > >
> > > > - What is the performance of the dataset on other families of models?
> > >
> > > We respectfully note that **we have extended the training experiments to Qwen3 in the revision**, which constitutes a distinct model family from Qwen2.5 in terms of both model architecture and pretraining data. Given limited time and computational resources, we selected Qwen3-4B as the most suitable candidate based on its model scale, relevance to the four reviewers' feedback, and strong community adoption [1], making it a representative and practical choice for evaluating generalization beyond the Qwen2.5 family. **We have also extended the LLM evaluation experiments to Qwen3, Ling, and Kimi-K2**, and we certainly look forward to evaluating the theorem-proving capabilities of broader model families in future work.
> > >
> > > > - The case study with detailed analysis, under DeepTheorem training
> > > > - RL-Zero does not increase the generation length
> > >
> > > We deeply appreciate the reviewer’s guidance toward these critical aspects of training dynamics. In direct response, **we have further revised the paper by adding a new subsection in Appendix E (Case Studies), focusing on generation lengths and error analyses**. Due to space constraints in this response, we summarize the key findings below and refer to the revised Appendix E for full cases:
> > >
> > > 1. **When training on DeepTheorem with RL-Zero, generation length drops sharply over the first few steps, and then gradually increases over the remaining training course**. This pattern is consistent across all backbone models (Qwen2.5 and Qwen3 of varying sizes) and data subsets (easy, mixed, hard). Furthermore, **the initial drop in response length is accompanied by a sharp rise in reward, suggesting that the model may be grasping at the task definition and trying to output minimal (often incorrect) answers to receive any reward signal. In contrast, in the later stage, more detailed reasoning becomes necessary to further increase the reward.**
> > >
> > > 2. **The predominant reasoning error in model-generated proof is overgeneralization**, i.e., drawing a universal conclusion directly from a few special cases. This is especially prevalent in the early stage of RL training and is mitigated as training proceeds, which may explain the short generation lengths in the beginning and the subsequent increase.
> > >
> > > 3. **SFT models exhibit frequent hallucination, whereas RL models do not**. We find that SFT models commonly modify conditions given in the question in their proof steps, whereas RL models rarely commit such errors. This observation aligns with previous findings on the susceptibility of SFT to hallucination [2].
> > >
> > > [1] According to the statistics provided by Hugging Face on December 2nd, 2025, the most popular Llama-4 model (Llama-4-Scout) has 207K downloads in the last month, followed by Llama-4-Maverick with 94K downloads. In contrast, Qwen3 0.6B, 4B, and 8B have 6.9M, 5.5M, 4.2M downloads in the last month, respectively.
> > >
> > > [2] Zhang et al. Siren's Song in the AI Ocean: A Survey on Hallucination in Large Language Models.

---

### Official Review · Reviewer_EGQ7 · 2025-11-08

**Soundness:** 2
**Presentation:** 3
**Contribution:** 3
**Rating:** 6
**Confidence:** 3

**Summary:**

The paper introduces DeepTheorem, a framework for informal theorem proving with large language models that combines a 121K-sample natural-language theorem dataset and reinforcement learning. By adapting RL-Zero to theorem proving with verifiable true/false rewards, the approach enhances reasoning beyond supervised fine-tuning. Evaluations on FIMO, Putnam, and HMMT benchmarks show that DeepTheorem-trained models outperform larger state-of-the-art systems in both proof accuracy and reasoning quality, highlighting the potential of natural-language-based reinforcement learning to advance LLM mathematical reasoning.

**Strengths:**

- Comprehensive experiments across multiple representative model scales.
- Introduction of a large, well-structured dataset for informal theorem proving.
- Innovative adaptation of RL-Zero to the informal theorem-proving setting.

**Weaknesses:**

- Many quality-control steps rely heavily on LLM judgments (e.g., contamination justification and difficulty annotation). A human evaluation of a sampled subset would strengthen confidence in the dataset’s integrity and annotation accuracy.
- The process evaluation methodology remains somewhat unclear (even with the prompt shown in the appendix). Compared to formal proof verification (e.g., in Lean), LLM-based evaluation is inherently less reliable. A more detailed comparison against prior process-evaluation models, including an analysis of reliability and typical failure modes, would be valuable.

Minor:
- Figure 2 (left): the formula before ‘is given by’ is not quite right, I guess something like ‘\pi / 2’ is missing.

**Questions:**

- How does the dataset compare to Omni-Math?

---

> ### Author Response · Authors · 2025-11-23
> **Response to Reviewer EGQ7**
>
> We sincerely thank the reviewer for recognizing the innovation and comprehensiveness of our work. We also appreciate the constructive comments and provide a point-by-point response below. Unless otherwise specified, all section numbers, line numbers, tables, and figures refer to the revised paper.
>
> ### 1. How does the dataset compare to Omni-Math?
>
> We thank the reviewer for raising this insightful question. **Omni-Math is a small-scale test set (4.4K samples) focused on short-answer questions, while DeepTheorem is designed for challenging, open-ended theorem proving**. Despite this fundamental difference, we acknowledge the valuable contribution of Omni-Math to the community, particularly its rigorous difficulty annotation framework. In fact, **as stated in Line 241-242, our difficulty pipeline explicitly adopts the methodology introduced in Omni-Math**, reflecting our commitment to building upon established best practices in the field.
>
> ### 2. Dataset quality control
>
> **The DeepTheorem dataset was constructed through a strict quality control pipeline that combines automated filtering with human-in-the-loop validation**. After each major processing step, we conducted manual inspections on random subsets to ensure fidelity and correctness.
>
> Moreover, the employment of LLMs for certain annotation steps is both well-documented and grounded in precedent. **For difficulty annotation, we adopt the same framework from the seminal work Omni-Math. For decontamination, we took inspiration from OpenMathInstruct-2 [1].** To increase transparency, we include examples of the three representative contamination types - identical generalizing, and logically converse - in Table 7 (formerly Table 5). Additionally, we have now also included examples from different difficulty levels in Table 6 to offer direct insight into the dataset's rigor and diversity.
>
> [1] OpenMathInstruct-2: Accelerating AI for Math with Massive Open-Source Instruction Data. https://arxiv.org/abs/2410.01560
>
> ### 3. Process evaluation methodology
>
> We fully acknowledge the reviewer’s concern regarding the reliability of LLM-based evaluation. To address this, we have substantially expanded our validation of the process evaluation methodology along three complementary aspects.
>
> First, instead of relying solely on API-based models, **we evaluated a third, open-source LLM - Ling-1T [2]**, one of the largest and most capable open-source models. The results are added to Table 10 in the appendix, which **exhibit a high consistency with the closed-source models**, with a correlation coefficient of more than 0.98 with both GPT-4o and o3-mini.
>
> Second, **we also incorporated a comparison with existing process evaluation models (Table 10)**, specifically Qwen2.5-Math-PRM [3], which scores every reasoning step in the model solutions (delimited by double line breaks) without requiring a dedicated prompt. **The results also align well with all three LLM judges, further verifying the robustness of our evaluation protocol**. The full correlation matrix is given below:
>
> |         | GPT-4o | o3-mini | Ling-1T | Q2.5-PRM-72B |
> | ------- | ------ | ------- | ------- | ------------ |
> | GPT-4o  | -      | 0.9545  | 0.9872  | 0.9964       |
> | o3-mini | -      | -       | 0.9866  | 0.9477       |
> | Ling-1T | -      | -       | -       | 0.9824       |
>
> Third, **we expanded the human evaluation** by recruiting 12 third-party expert annotators (all graduate students majoring in mathematics) to perform pairwise comparisons of model responses across 100 questions and 10 models. **The results (Table 11) are highly consistent with preferences computed from the scores of LLM judges, demonstrating strong external validity**. We will release these human annotations along with other artifacts created in this research.
>
> [2] Every Activation Boosted: Scaling General Reasoner to 1 Trillion Open Language Foundation. https://arxiv.org/abs/2510.22115
>
> [3] The Lessons of Developing Process Reward Models in Mathematical Reasoning. https://aclanthology.org/2025.findings-acl.547/
>
> ### 4. Incomplete Figure 2
>
> The reviewer is correct that the question in Figure 2 is not complete. This was an error during the paper's typesetting, and we have fixed it in the revision. We thank you for your keen eye to detail.
>
> Again, we sincerely thank the reviewer's insightful and constructive feedback. We hope the responses have addressed the concerns.

---

### Author Response · Authors · 2025-12-02
**General Response and Revision Summary**

We sincerely thank all reviewers for their time and effort in reviewing our submission. We are grateful that our method is recognized as _novel_ (Reviewer xMBg), _innovative_ (EGQ7), and _empirically effective_ (xMBg, zq5f, Ua9x); that our data resource is acknowledged as _large-scale_ (EGQ7, xMBg, zq5f, Ua9x), _carefully annotated_ (zq5f), _sound_ (xMBg), and _valuable to the field_ (xMBg); that our experiments are deemed _comprehensive_ (EGQ7) and our writing _easy to follow_ (zq5f).

We also deeply appreciate the reviewers' thoughtful, constructive, and valuable critiques and questions, which have been comprehensively addressed and significantly helped us strengthen the paper. As the discussion phase draws to an end, we would like to summarize the key revisions we have made in response to the feedback. All references to sections, tables, and figures refer to the revised manuscript.

## Improved positioning, clarity, and readability

- To stress the scope and positioning of our work within informal theorem proving, we (1) minimized discussion of formal theorem proving datasets and methods, (2) enhanced discussion of and comparison to other informal theorem datasets, and (3) removed the potentially misleading DeepSeek-Prover results from the main experiments.

- To enhance readability and presentation quality, we (1) replaced all figures with vector graphs, (2) fixed typesetting errors, (3) improved layout in Figure 7, and (4) added a complete topic breakdown in Table 5 to supplement Figure 3.

- To clarify the quality control of our dataset construction pipeline, we highlighted that (1) our annotation methodologies for both difficulty and contamination are rigorous and follow well-established precedence in the literature, (2) the generation of theorem variants are unambiguously grounded in mathematical logic, (3) human inspection is performed in every processing step, and (4) all intermediate annotations is fully open-sourced and publicly available. We also (5) added Table 6, showcasing data samples of different difficulty levels.

## Improved technical soundness

- To further justify our LLM-based process evaluation (Appendix D), we (1) incorporated a third, open-source LLM judge, (2) added comparisons with a dedicated process reward mdoel (Qwen2.5-Math-PRM-72B), (3) emphasized the highly consistent inter-judge correlation across all four models, and (4) expanded human evaluation to 12 mathematical experts across 100 questions and 10 models, showing strong alignment with model scores.

- To expand the comprehensiveness of our experiments (Appendix C), we (1) incorporated three SOTA models to LLM evaluations in Table 4, (2) included new training experiments using Qwen3 backbone, (3) conducted a series of ablation studies with controlled dataset sizes and difficulty compositions, and (4) highlighted new insights from these additional experiments.

- To deepen the analysis of training dynamics (Appendix E), we (1) extended the case studies, (2) analyzed how and why response length evolves during RL training, and (3) systematically analyzed reasoning error types of the trained models.

Once again, we sincerely thank the reviewers, area chairs, and organizers for their time, effort, and invaluable contribution to improving the soundness, clarity, and overall quality of our paper.

---

### Meta-Review · Area_Chair_H8sp · 2026-01-06

**Summary:**

The paper proposes a new framework for informal theorem proving, including a large-scale training dataset, a method, a benchmark, and an evaluation protocol. Reviewers agree that the training data is large and can substantially improve model performance.

The main concern is that both dataset construction and evaluation rely entirely on LLMs. Reviewers 1, 3, and 4 argue that LLM-based evaluation cannot guarantee semantic correctness, even when inter-judge agreement is high. Reviewers 3 and 4 also raise concerns about data contamination. Reviewers 1 and 3 question the quality control of the LLM-generated data, with Reviewer 3 further noting that the dataset appears to be a form of distillation from o3-mini.

Additional concerns relate to the experimental setup. Reviewers 2 and 4 note that the experimental analysis is insufficient and the compared methods are outdated.

The authors’ rebuttal partially addresses some issues. The authors clarify that human inspection was used to validate the training data and that the evaluation variants were constructed by humans to avoid contamination. However, several core concerns remain unresolved. In particular, alignment between human and LLM evaluations is imperfect, with over half of the models’ rankings are slightly different, and the training data is not fully decontaminated. As a result, it is unclear whether the dataset will benefit current state-of-the-art provers.

Given the convergence of concerns across reviews and the mostly negative ratings, I lean toward rejection.

**Reviewer Concerns:**

Concerns about the experimental analysis and evaluation data contamination are likely to be addressed.

Concerns regarding the use of LLMs as judges and the lack of comparisons with more state-of-the-art models are not fully resolved.

**Reviewer Scores:**

Reviewer 2 may raise the score, as most of the concerns are addressed in the rebuttal.

Reviewers 3 and 4 are unlikely to raise their ratings, as their major concerns remain unresolved.

---

### Decision · Program_Chairs · 2026-01-26

Reject